# GRAM: A Generative Foundation Reward Model
# for Reward Generalization

**Chenglong Wang** [1]   **Yang Gan** [1]   **Yifu Huo** [1]   **Yongyu Mu** [1]   **Qiaozhi He** [1]   **Murun Yang** [1]   **Bei Li** [2]   **Tong Xiao** [1 3]
**Chunliang Zhang** [1]   **Tongran Liu** [4]   **Jingbo Zhu** [1 3]

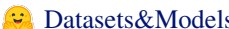 Code         🤗 Datasets&Models

## Abstract

In aligning large language models (LLMs), reward models have played an important role, but are standardly trained as discriminative models and rely only on labeled human preference data. In this paper, we explore methods that train reward models using both unlabeled and labeled data. Building on the generative models in LLMs, we develop a generative reward model that is first trained via large-scale unsupervised learning and then fine-tuned via supervised learning. We also show that by using label smoothing, we are in fact optimizing a regularized pairwise ranking loss. This result, in turn, provides a new view of training reward models, which links generative models and discriminative models under the same class of training objectives. The outcome of these techniques is a foundation reward model, which can be applied to a wide range of tasks with little or no further fine-tuning effort. Extensive experiments show that this model generalizes well across several tasks, including response ranking, reinforcement learning from human feedback, and task adaptation with fine-tuning, achieving significant performance improvements over several strong baseline models.

## 1. Introduction

Reward models are a fundamental concept in reinforcement learning and define what an agent optimizes for. For large language models (LLMs), fine-tuning with reward models is a common post-training step to align the model outputs with desired behaviors and objectives. A widely adopted approach is to learn reward models that capture human preferences and fine-tune the LLMs to generate outputs that align with these preferences. Reinforcement learning from human feedback (RLHF) is an early example of such approaches (Christiano et al., 2017; Stiennon et al., 2020). Now, work in this area is underway more broadly. One recent example is a series of models by OpenAI (2024), in which human-like thinking and complex reasoning can be achieved through large-scale reinforcement learning.

While quite successful, reward models are costly to apply. This is in part because of the complexity of reinforcement learning algorithms and in part because of the difficulty in annotating training data. There has been much work on simplifying the use of reward models and improving alignment efficiency. One strand of research explores more direct ways to align LLMs with human feedback, employing either supervised fine-tuning methods (Rafailov et al., 2023; Touvron et al., 2023) or inference-time alignment methods (Lee et al., 2021). Another strand of research focuses on replacing human feedback with AI-generated feedback, which is cheaper to obtain (Dubois et al., 2023; Lee et al., 2024).

However, although applying reward models to LLMs is a compelling direction, training these models still relies heavily on labeled data. For example, we generally need to collect or create a significant amount of task-specific human preference data and optimize the models with considerable training effort (Stiennon et al., 2020; Xu et al., 2024). If we think about the problem a bit closer from the LLM perspective, we might expect that reward models can be trained on unlabeled data in such a way as to produce a single pre-trained reward model that can be easily adapted to tasks of interest. This would change the way we align LLMs: we can pre-train a foundation model that assembles a broad general knowledge of how to reward, and a single such pre-trained model can be deployed for many particular rewarding tasks with only small costs of further fine-tuning or prompting.

This idea is appealing but challenging. The difficulty arises from the fact that the systems cannot directly generate their own supervision signals from text for training reward models, as self-supervision methods do. One approach is to

---

[1] School of Computer Science and Engineering, Northeastern University, Shenyang, China [2] Meituan Inc. [3] NiuTrans Research, Shenyang, China [4] CAS Key Laboratory of Behavioral Science, Institute of Psychology, CAS, Beijing, China. Correspondence to: Chenglong Wang <clwang1119@gmail.com>, Tong Xiao <xiaotong@mail.neu.edu.cn>.

*Proceedings of the 42nd International Conference on Machine Learning*, Vancouver, Canada. PMLR 267, 2025. Copyright 2025 by the author(s).

collect large-scale preference data for general use and train a reward model on this data to improve generalization ([Cui et al., 2023](#); [Liu et al., 2024](#)). However, in this case, large amounts of unlabeled data are still largely overlooked.

In this paper, we propose a solution to this problem that learns reward models not only from human-annotated preference data but also from unlabeled data. To do this, we develop a generative model that can predict, given the input and a pair of responses, which one is better. The training of this model involves two stages. In the first stage, we pre-train the model on input-response data to learn the correspondence between inputs and responses. This process does not require preference-annotated data and so can be easily scaled up to gain more general knowledge of response comparison. In the second stage, we fine-tune the model using human preference data to predict the preference between two responses. The resulting foundation reward model can be directly applied to downstream tasks, such as policy training, or further fine-tuned with a small amount of task-specific data.

To make the model generalize better, we incorporate label smoothing into reward model training. We show that the training objective can be reformulated into a nice form: we are essentially optimizing the Bradley-Terry loss ([Bradley & Terry, 1952](#)) under the condition of label smoothing. This result is elegant, as it unifies generative and discriminative models in reward modeling to some extent. Though label smoothing seems not so popular in the development of recent LLMs, it turns out to be very beneficial for training generative reward models.

The foundation reward models can be applied to a wide range of tasks. In our experiments, we test it in three different settings: response ranking, RLHF, and adaptation. Our model demonstrates strong generalization results across all test cases with little or no fine-tuning effort and improves performance significantly compared with various discriminative and generative baseline models. Notably, when training reward models with the LLaMA-3.1-8B-Instruct model, our model achieves gains of 11.0 and 5.1 points over vanilla discriminative and generative reward models, respectively, on the average accuracy of RewardBench.

## 2. Preliminaries

In this section, we outline some basic concepts and notations of reward modeling.

### 2.1. Training Reward Models

In the LLM literature, a reward model is typically written as a function $r_\phi(x, y)$, where $\phi$ is the set of model parameters, $x$ is the input, and $y$ is the response. Throughout this work, an "input" can be an arbitrary token sequence fed into an

Minimizing the Bradley-Terry loss (pairwise ranking loss):
$$-\log(\sigma(r_\phi(x, y_a) - r_\phi(x, y_b)))$$

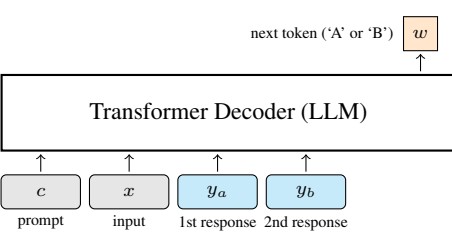

(a) Discriminative Models (Trained as Classifiers)

Minimizing the negative probability of token prediction:
$$-\log \pi_\phi(w = \mathrm{A}|[c, x, y_a, y_b])$$

(b) Generative Models (Trained as LLMs)

Figure 1: Architectures of discriminative and generative reward models. In discriminative models, the reward model is a scoring function that is trained to minimize the pairwise ranking loss between two responses. In generative models, we use an LLM to predict the label token given a prompt, an input, and a pair of responses. This model can be trained in the same way as standard LLMs.

LLM, such as *What is the capital of France?*, and a "response" is the token sequence produced by the LLM as a result of that input, such as *Paris*.

A widely used architecture of such functions is a Transformer decoder stacked without a Softmax layer on top, as illustrated in Figure 1 (a). This model can be viewed as a discriminative classification model, and is commonly trained using the Bradley-Terry loss, given by

$$\mathcal{L}_\mathrm{d} = -\mathbb{E}_{(x, y_a, y_b) \sim D_r}$$
$$[\log(\sigma(r_\phi(x, y_a) - r_\phi(x, y_b)))] \quad (1)$$

where $D_r$ is the training dataset consisting of tuples of input $x$ and response pair $(y_a, y_b)$ with the preference $y_a \succ y_b$. While this loss function considers pairwise ranking between responses, the trained reward model is used as a scoring function that assigns a numerical score $r_\phi(x, y)$ to any response $y$, together with the corresponding input $x$.

Reward models can also be generative models ([Zhang et al., 2024](#); [Shiwen et al., 2024](#)). In this case, we can simply use an LLM as a reward model, as illustrated in Figure 1 (b). This model works as follows. First, we input a prompt $c$,

along with the tuple $(x, y_a, y_b)$, to the LLM. The prompt is a text describing the task. For example,

> *You are given two responses to a user input. Evaluate which response is better based on quality, relevance, and clarity. If the first response is better, return 'A'. If the second response is better, return 'B'.*

Then, the LLM predicts subsequent tokens based on this input sequence. Let $w$ be the token predicted by the LLM. If $w = $ A, it indicates a preference for $y_a$ over $y_b$; if $w = $ B, then $y_b$ is preferred.

The loss function can be defined as the log-probability of predicting 'A':

$$\mathcal{L}_g = -\mathbb{E}_{(c,x,y_a,y_b) \sim D_r}[\log \pi_\phi(w = \text{A}|s)] \quad (2)$$

where $s$ denotes the string $[c, x, y_a, y_b]$[1], and $\pi_\phi(\cdot)$ denotes the probability of token prediction by the LLM.

When applying this model to score a new input-response pair $(x', y')$, we generate a reference response $y_{\text{ref}}$ by using the LLM, and concatenate $x'$, $y'$ and $y_{\text{ref}}$ into $s' = [c, x', y', y_{\text{ref}}]$. Additionally, to mitigate the positional bias problem (Wang et al., 2023), we introduce an alternative input order by transposing the positions of responses, i.e., presenting $y_{\text{ref}}$ before $y'$, to construct a secondary input string $s'_T = [c, x', y_{\text{ref}}, y']$. The reward for $(x', y')$ is thus defined as the log-probability that $y'$ is preferred over $y_{\text{ref}}$:

$$r_\phi(x', y') = \frac{\pi_\phi(w = \text{A}|s') + \pi_\phi(w = \text{B}|s'_T)}{2} \quad (3)$$

where the reward score ranges from 0 to 1.

### 2.2. Applying Reward Models

Three applications of foundation reward models can be considered in LLMs. One simple application is response ranking, where a number of responses are given, and we score and rank these responses. This approach is often used in reranking the LLM outputs. For example, in best-of-$n$ sampling, we select the best output from the top $n$ candidate outputs via a reward model (Lee et al., 2021; Fernandes et al., 2022; Gao et al., 2023).

A second application is reward-based fine-tuning, where the reward model provides feedback to optimize the LLM. For example, in RLHF, a reward model is used in proximal policy optimization (PPO) (Wang et al., 2022) to fine-tune the LLM for better alignment with human preferences (Ouyang et al., 2022; Bai et al., 2022).

A third application is reward model adaptation. If we have labeled human preference data for a task, we can fine-tune

---

[1]In this work, we will use $s$ interchangeably to refer to either the tuple of a training sample or a string representing that tuple.

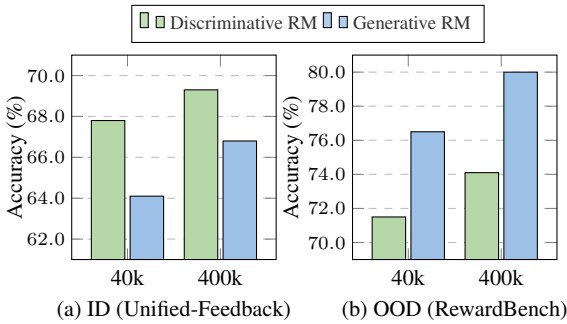

Figure 2: Accuracies of discriminative and generative reward models on the ID and OOD test sets.

the reward model further to better adapt it to the task. The fine-tuned reward model can then be applied to LLM fine-tuning as usual.

## 3. A Generative Foundation Reward Model

In this section we describe a **G**enerative foundation **R**ew**a**rd **M**odel, called GRAM.

### 3.1. Why Generative Models

Both discriminative and generative models have been widely adopted in reward modeling, but we found that generative models were more generalizable and better suited to our work. To study this issue, we trained both types of models on a subset of 400k and 40k samples from the Unified-Feedback dataset[2]. We then evaluated these models on an in-distribution (ID) test set, consisting of 1k test samples from Unified-Feedback, and an out-of-distribution (OOD) test set, consisting of 3k samples from RewardBench (Lambert et al., 2024). As shown in Figure 2, the discriminative model is better on the ID test data, while the generative model is better on the OOD test data. While preliminary, this result supports the finding in previous work by Yang et al. (2024) and Zhang et al. (2024): LLMs can generalize better for reward modeling.

In fact, discriminative and generative models have many similar design choices, such as using Transformer decoders (i.e., LLMs) to encode input-response pairs, but the training strategies make them behave quite differently. Training with the pairwise ranking loss provides a strong supervision signal. Given that the reward model is a well-trained LLM, it is more prone to overfit when simply mapping an input-response pair to the reward model. Generative models are essentially trained in a similar manner, but with much noisier samples. For example, each time, we need to model two responses in the same sequence, and adding an extra prompt to the sequence introduces more modeling challenges. The

---

[2]https://huggingface.co/datasets/llm-blender/Unified-Feedback

diversity and variability of the samples make the training task more difficult, which in turn encourages the model to generalize more.

Furthermore, recent research highlights the superior flexibility of generative models compared to discriminative models in their adaptability to various LLM enhancement techniques. For example, the seamless integration of chain-of-thought reasoning within a generative reward model has been shown to improve reward accuracy (Mahan et al., 2024). Beyond merely adopting such reasoning patterns, the generative reward model can be designed to perform long-form reasoning before generating preferences (Chen et al., 2025; Guo et al., 2025). This enhanced flexibility makes the generative model an ideal choice for our work, as we aim to build a more versatile foundation reward model.

### 3.2. A Two-stage Training Method

Unlike previous work, we do not use only human preference data to train reward models. Instead, we train them using an unsupervised pre-training task and a supervised fine-tuning task, as described in this section.

#### 3.2.1. TASK 1: PRE-TRAINING

Intuitively, one might expect that an LLM-based reward model is capable enough to model the inputs, responses, and correspondences between them, as LLMs have been trained on huge amounts of text. Unfortunately, modeling the input-response correspondence is not covered by standard pre-training and fine-tuning tasks for LLMs, and the model still needs to adapt to this modeling task. This has been demonstrated by recent work, where the understanding of responses is found to be very important in modeling human preferences (Wang et al., 2024e; Zheng et al., 2023).

To improve response understanding, we train the LLM to generate responses given the input, as illustrated in Figure 3 (a). We refer to this as a "pre-training" procedure, as it does not require human preference data, although it differs from standard pre-training methods in LLMs. Let $D_u$ be a dataset consisting of tuples of input and pairs of responses with no preference specified. This loss function can be defined as:

$$\mathcal{L}_{\text{pre}} \quad = \quad -\mathbb{E}_{(x,y_a,y_b)\sim D_u}\left[\log \pi_\phi([y_a, y_b]|x)\right] \quad (4)$$

Here $y_a$ and $y_b$ can be generated by an LLM. So, building such a dataset is straightforward.

This training objective is similar to that used in instruction fine-tuning (Ouyang et al., 2022). The difference between our method and instruction fine-tuning is that we train the LLM to generate two responses, while instruction fine-tuning trains the LLM to produce a single correct response to the input. By learning the mapping from inputs to varied responses, the model can better understand the responses.

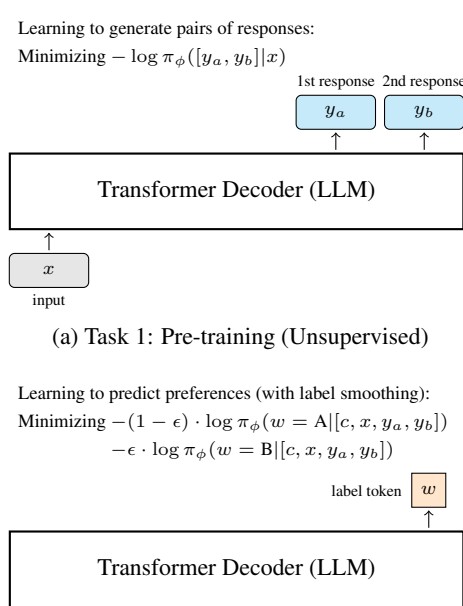

(a) Task 1: Pre-training (Unsupervised)

(b) Task 2: Fine-tuning (Supervised)

Figure 3: Illustration of the two-stage training method. In the first stage, we pre-train the model via response generation, which is an unsupervised task. In the second stage, we fine-tune the model to generate preferences in a standard supervised manner.

Furthermore, since the two responses are generated at the same time, the model can gain some general knowledge of response comparison. Note that the order of responses does not matter in pre-training, so we can swap them to create more diverse samples for robust training.

#### 3.2.2. TASK 2: FINE-TUNING

The goal of fine-tuning here is to adapt the model to predict preferences between responses, which is not directly captured by pre-training. We can do this by using the same method described in Eq. (2) (see Figure 3 (b)). As pre-training has helped the model gain knowledge of response comparison from relatively large amounts of data, fine-tuning is easier and requires much less data. In this work, we consider using general-purpose preference data to fine-tune our pre-trained model, thereby obtaining a foundation model that works in various rewarding tasks. If users have their own data, such as human-annotated preference data for a specific task, they can fine-tune this model further.

### 3.3. Training with Label Smoothing

To further improve generalization, we incorporate fine-tuning with label smoothing. The idea of label smoothing is to redistribute the probability mass across all predicted

tokens by distracting a fraction of the probability from the correct token (denoted by $w^*$) to the incorrect tokens. For fine-tuning with label smoothing, the loss for a sample $s$ can be defined as:

$$
\begin{aligned}
\mathcal{L}_{\mathrm{ls}}(s) \;=\; & -\mathbb{E}_{w \sim |V_l|}\Big[ \\
& (1-\epsilon) \cdot \mathbf{1}\{w = w^*\} \log \pi_\phi(w|s) \\
& + \frac{\epsilon}{|V_l| - 1} \cdot \mathbf{1}\{w \neq w^*\} \log \pi_\phi(w|s)\Big] \quad (5)
\end{aligned}
$$

where $\epsilon$ is the smoothing factor, and $V_l$ is the vocabulary of label tokens. This formula is general and can handle multi-class problems (i.e., $|V_l| > 2$). But, to simplify the discussion, we still restrict ourselves to the case of two classes. Hence, we express the loss as

$$
\begin{aligned}
\mathcal{L}_{\mathrm{ls}}(s) \;=\; & -\Big[(1-\epsilon) \cdot \log \pi_\phi(w = \mathrm{A}|s) \\
& + \epsilon \cdot \log \pi_\phi(w = \mathrm{B}|s)\Big] \quad (6)
\end{aligned}
$$

We can rewrite this formula into another form by noting that $\pi_\phi(w|s)$ is the output of a Softmax layer. This gives

$$
\begin{aligned}
\mathcal{L}_{\mathrm{ls}}(s) \;=\; & -\Big[(1-\epsilon) \cdot \log \frac{e^{Z_a(s)}}{e^{Z_a(s)} + e^{Z_b(s)}} \\
& + \epsilon \cdot \log \frac{e^{Z_b(s)}}{e^{Z_a(s)} + e^{Z_b(s)}}\Big] \quad (7)
\end{aligned}
$$

where $Z_a(s)$ and $Z_b(s)$ are the logits input to the Softmax function.

Using simple algebra, we obtain:

$$
\mathcal{L}_{\mathrm{ls}}(s) \;=\; \underbrace{-\log \sigma(Z_a(s) - Z_b(s))}_{\text{Bradley-Terry model}} + \underbrace{\epsilon \cdot (Z_a(s) - Z_b(s))}_{\text{regularization term}} \quad (8)
$$

See Appendix D.2 for a more detailed derivation. The first term of Eq. (8) is the loss based on the Bradley-Terry model, and the second term serves as a regularization term. This result is quite interesting. We are actually optimizing a regularized Bradley-Terry loss. It also establishes a connection between the discriminative and generative training methods discussed in Section 2.1: both methods essentially train LLMs to perform pairwise ranking.

Note that while label smoothing is theoretically appealing (Müller et al., 2019; Lukasik et al., 2020), past experience shows that it is not very helpful for LLMs. One recent example of this problem is the experiments by Ruan et al. (2024), in which label smoothing degrades the performance of LLMs in some cases. By contrast, in our work, this technique turns out to be very beneficial. We will see in Figure 10 that using label smoothing is critical in training our reward models.

## 4. Experiments

We evaluated GRAM on various applications, including its accuracy on the response ranking, effectiveness in reward-based fine-tuning, and adaptability to rewarding tasks.

### 4.1. Setups

We initialized our GRAM model with the LLaMA-3.1-8B-Instruct and LLaMA-3.2-3B-Instruct models, using a subset of 400k samples from Unified-Feedback for each. The learning rates were set to 2e-5 for the first stage and 1e-5 for the second stage, with training conducted over one epoch in each stage. Note that while this data includes preference labels, we do not use these labels in our pre-training process. Instead, we only use the response pairs to simulate unlabeled data and validate the effectiveness of our method. In the second stage, the label smoothing parameter was set to 0.1, with other settings tested as shown in Figure 10. More details can be found in Appendix B.

### 4.2. Baselines

We compared GRAM with several strong baselines: *LLM-as-a-Judge*, where we prompted LLMs like GPT-4o to generate preferences; *open-source reward models*, open-source discriminative and generative reward models that approximate 3B or 8B, including ArmoRM-Llama3-8B-v0.1 (Wang et al., 2024d), and others; and *training on the same preference dataset*, denoting the standard reward models trained on discriminative and generative frameworks using Unified-Feedback, respectively (Discriminative RM and Generative RM). We also compared GRAM with several approaches designed to enhance generalization. These include the standard *Label Smoothing*, which prevents the model from becoming overly confident in its predictions; *Freeze*, which fixes certain parameters of the LLM during training (Zilly et al., 2021); and *Regularization*, which adds the discriminative reward model loss using the SFT loss function (Yang et al., 2024).

### 4.3. Pair-wise Response Ranking

**Task Setups.** The pair-wise ranking is commonly used to test reward models (Lambert et al., 2024). Given test data $D_{\mathrm{pair}}^t = \{(x^t, y_a^t, y_b^t)\}$, where $x^t$ denotes the test input, and $y_a^t$ and $y_b^t$ denote its corresponding responses, the task is to identify the preferred response. The test sample $(x^t, y_a^t, y_b^t)$ can be evaluated using a reward model:

$$
\mathrm{Rank}(y_a^t, y_b^t) = \begin{cases} y_a^t \succ y_b^t, & \text{if } r_\phi(x^t, y_a^t) > r_\phi(x^t, y_b^t) \\ y_a^t \prec y_b^t, & \text{if } r_\phi(x^t, y_b^t) > r_\phi(x^t, y_a^t) \\ \text{Tie}, & \text{if } r_\phi(x^t, y_a^t) = r_\phi(x^t, y_b^t) \end{cases} \quad (9)
$$

For generative reward models, when calculating the reward score for one of two responses, the other is used as the

| Method | UNIFEED | REWARDBENCH | | | | | HHH-ALIGNMENT | | | |
| --- | --- | --- | --- | --- | --- | --- | --- | --- | --- | --- |
| | | Chat | Chat-Hard | Safety | Reasoning | Avg. | Harmless. | Helpful. | Honest. | Avg. |
| **_LLM-as-a-Judge_** | | | | | | | | | | |
| GPT-4[†] | - | 95.3 | 74.3 | 87.6 | 86.9 | 86.0 | 96.6 | 91.5 | 82.0 | 90.0 |
| GPT-4o[†] | - | 96.6 | 70.4 | 86.5 | 84.9 | 84.6 | 98.3 | 90.0 | 83.6 | 90.6 |
| GPT-3.5-turbo[†] | - | 92.2 | 44.5 | 62.3 | 59.1 | 64.5 | 74.1 | 78.0 | 72.1 | 74.7 |
| **_Open-Source Reward Models_** | | | | | | | | | | |
| pair-preference-model-LLaMA3-8B[†] | - | 96.9 | 76.8 | 90.5 | 97.3 | 90.4 | 90.2 | 86.2 | 76.7 | 84.4 |
| GPM-Gemma-2B[†] | - | 71.5 | 69.7 | 81.2 | 75.5 | 74.5 | 76.3 | 68.4 | 66.0 | 70.2 |
| Selene-1-Mini-Llama-3.1-8B[†] | - | 93.6 | 79.4 | 89.3 | 94.3 | 89.1 | 90.6 | 87.5 | 78.4 | 85.5 |
| Skywork-Critic-Llama-3.1-8B[†] | - | 93.6 | 81.4 | 91.1 | 89.8 | 89.0 | 91.3 | 87.1 | 76.2 | 84.9 |
| **_Training on the Same Preference Dataset (LLaMA-3.1-8B-Instruct)_** | | | | | | | | | | |
| Discriminative RM (Baseline) | 69.3 | 80.7 | 73.5 | 74.9 | 67.4 | 74.1 | 75.3 | 78.6 | 74.1 | 76.0 |
| Discriminative RM + Freeze | 66.6 | 81.3 | 75.2 | 78.8 | 64.2 | 74.9 | 74.1 | 81.4 | 77.0 | 77.5 |
| Discriminative RM + Regularization | **72.7** | 85.8 | 74.3 | 80.1 | 69.5 | 77.4 | 74.6 | 84.1 | 80.3 | 79.7 |
| Generative RM (Baseline) | 66.8 | 88.6 | 79.3 | 79.6 | 72.4 | 80.0 | 76.9 | 83.9 | 80.2 | 80.3 |
| Generative RM + Freeze | 65.4 | **91.2** | 82.3 | 81.7 | 75.1 | 82.6 | 77.2 | 81.9 | 81.3 | 80.1 |
| Generative RM + Label Smoothing | 67.9 | 89.5 | 80.1 | 80.4 | 72.6 | 80.7 | 77.0 | 80.2 | 81.2 | 79.5 |
| GRAM (Ours) | 70.4 | 90.4 | **86.9** | **84.6** | **78.3** | **85.1** | **83.9** | **88.6** | **83.7** | **85.4** |
| **_Training on the Same Preference Dataset (LLaMA-3.2-3B-Instruct)_** | | | | | | | | | | |
| Discriminative RM (Baseline) | 68.3 | 86.6 | 71.6 | 71.9 | 61.1 | 72.8 | 79.8 | 77.9 | 70.5 | 76.1 |
| Discriminative RM + Freeze | 63.0 | 83.9 | 67.4 | 73.5 | 57.0 | 70.5 | 63.5 | 69.1 | 69.8 | 67.5 |
| Discriminative RM + Regularization | 65.6 | 85.4 | 68.7 | 74.2 | 56.5 | 71.2 | 73.4 | 72.1 | 71.7 | 72.4 |
| Generative RM (Baseline) | 65.3 | 87.9 | 78.8 | 77.5 | 71.1 | 78.8 | 75.5 | 77.3 | 78.8 | 77.2 |
| Generative RM + Freeze | 63.7 | 83.6 | 76.5 | 78.3 | 68.8 | 76.8 | 73.2 | 76.1 | 77.0 | 75.4 |
| Generative RM + Label Smoothing | 66.2 | 85.7 | 78.1 | **80.3** | 72.0 | 79.0 | 76.4 | 78.3 | 79.7 | 78.1 |
| GRAM (Ours) | **70.6** | **90.6** | **83.9** | 79.8 | **80.2** | **83.6** | **81.2** | **84.1** | **80.3** | **81.9** |

Table 1: Accuracies (%) on the pair-wise ranking with both ID (UNIFEED) and OOD (REWARDBENCH and HHH-ALIGNMENT) test sets. The best performance in each group is in **bold** and the second best one is underlined. Results marked with [†] for RewardBench are from Lambert et al. (2024). The other baseline results are obtained by testing this available model or API. We use a dotted line to distinguish between the discriminative and generative reward models. We report the average accuracy for RewardBench and HHH-Alignment sets in the "Avg." column.

reference response. For example, to compute $r_\phi(x^t, y_a^t)$, we use $y_b^t$ as the reference response.

**Results on Generalization.** We used a pair-wise response ranking task to evaluate the generalization capability of GRAM. Table 1 shows the results of GRAM and its baselines on both ID and OOD test sets. Firstly, the results here confirm the findings from Section 3.1, demonstrating that discriminative reward models are less effective than generative reward models in generalization, even when enhanced generalization methods are applied. Interestingly, GRAM also outperforms the discriminative reward model on the ID test set, underscoring the substantial improvement and generalization capability of GRAM in reward modeling. Furthermore, compared to LLM-as-a-Judge methods, 8B Generative RM (Baseline) achieves a competitive score, while GRAM shows a notable improvement, increasing the

average score on the RewardBench from 80.0 to 85.1. This shows that relying only on prompt engineering in a suboptimal reward model, even with a strong LLM, is insufficient. This finding here is consistent with the result in Zhang et al. (2024). Additionally, compared to open-source reward models trained on large-scale, high-quality labeled data, GRAM demonstrates competitive performance. As shown in Figure 6, GRAM outperforms these open-source models as more fine-tuning data is used, achieving an average accuracy of 91.6 on the RewardBench.

From the results, we also observe that GRAM underperforms compared to discriminative models on ID data, which may raise concerns about overfitting in the discriminative models rather than better generalization. However, this is not the case. First, the ID test set evaluates the model's ability to learn human preferences from labeled data, and our goal is to excel in both ID and OOD tasks. As shown by the

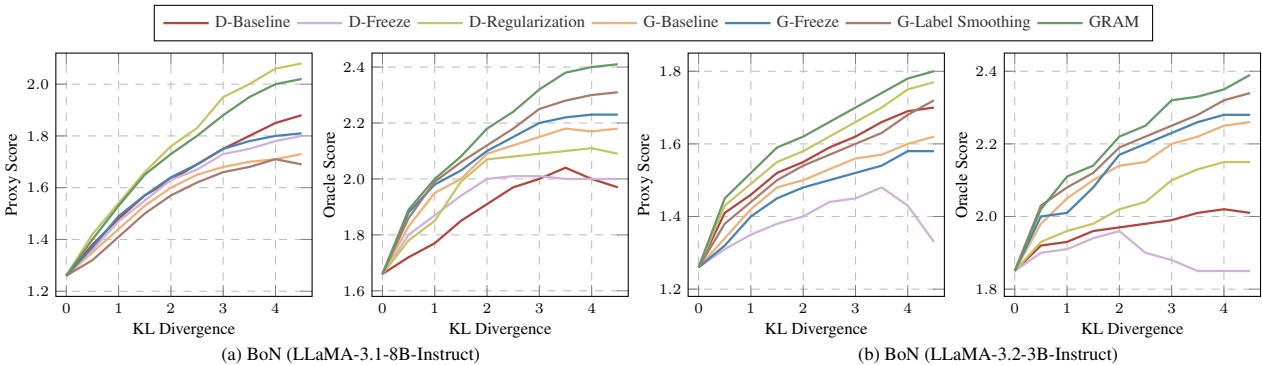

Figure 4: Performance of GRAM and its baselines on BoN sampling. We use proxy scores to assess preference learning and oracle scores to evaluate the generalization capability. "D-" and "G-" denote that the reward model is trained using discriminative and generative reward modeling frameworks.

LLaMA-3.1-8B-Instruct results in Table 1, GRAM achieves the best OOD results and second-best ID results. Second, while GRAM underperforms relative to Discriminative RM+Regularization on the LLaMA-3.1-8B-Instruct model, it outperforms both the Discriminative RM (Baseline) and Discriminative RM+Freeze, demonstrating GRAM's strong performance. Additionally, we find that regularization's effectiveness is model-dependent, as it performs worse than GRAM on the LLaMA-3.2-3B-Instruct model.

## 4.4. List-wise Response Ranking

In practice, multiple responses are typically generated for reranking. Given a list-wise test set $D_{\text{list}}^t = \{(x^t, y_1^t, y_2^t, \cdots, y_k^t)\}$, where $k$ denotes the list size, we begin by randomly selecting a response $y_j^t$ as the reference response $y_{\text{ref}}$. We then compute reward scores $\{r_\phi(x^t, y_1^t), r_\phi(x^t, y_2^t), \cdots, r_\phi(x^t, y_{k-1}^t)\}$ for the remaining responses via Eq. 3. These scores are subsequently used for ranking these responses[3]. Additionally, when the goal is to find the best response from the response list, a straightforward linear search approach can be employed. Specifically, we start by defining $y_1^t$ as the best response $y_b^t$ and comparing it iteratively with the remaining responses with the generative reward model. At each comparison, if $y_b^t$ is found to be inferior, it is replaced by the compared response. Through this process, we can determine the best response. To support parallel computation and enhance efficiency, we also incorporate optimization algorithms, such as divide-and-conquer.

**Task Setups.** We used best-of-$n$ (BoN) sampling to evaluate GRAM on list-wise ranking. We performed BoN sampling on the LLaMA-3.1-8B-Instruct model using $k$ responses per input. The test set was AlpacaEval2 (Li et al., 2023). In all BoN experiments, we trained a proxy reward

model on a 40k subset of the Unified-Feedback dataset to provide a proxy score for the responses selected by GRAM and its baselines. Additionally, we trained an oracle reward model using preference data from AlpacaFarm (Dubois et al., 2023), which accurately measures response quality to assess generalization, as AlpacaFarm's preference data is distributed alongside AlpacaEval2. Following Gao et al. (2023)'s work, we varied the KL Divergence between 0 and 4.5, which corresponds to a range of $k$ from 1 to 244 responses, according to the equation $\text{KL}_{\text{BoN}} = \log k - \frac{k-1}{k}$.

**Results of Best-of-$n$ Sampling.** Figure 4 presents the BoN sampling results for reward models of the 3B and 8B sizes. When comparing discriminative and generative reward models, we observe that the discriminative reward model yields a strong proxy score but underperforms in oracle scores. This indicates that while the discriminative reward model exhibits robust preference learning, its generalization capability is weaker, consistent with observations in pair-wise ranking. In contrast, GRAM excels in list-wise ranking in both proxy and oracle reward model evaluations. We observe a decline in oracle scores for baseline models when the KL divergence exceeds 3, attributable to over-optimization. However, GRAM mitigates this issue, demonstrating its potential as a reliable foundation reward model in RLHF. We also further evaluate its performance during PPO fine-tuning, as shown in Appendix C.1.

## 4.5. Reward Model Adaptation

The adaptability of a reward model is crucial for its performance across various tasks, as it enables the model to effectively adjust to different environments and preferences (Cheng et al., 2023; Wang et al., 2024a). To evaluate GRAM's adaptability, we conducted experiments in two distinct tasks: adapting to the summarization task and adapting to the harmlessness preference type. For each task, we fine-tuned GRAM on a small, labeled dataset containing

---

[3]In the list-wise response ranking process, the reward score for the reference response $y_k^t$ is set to 0.5.

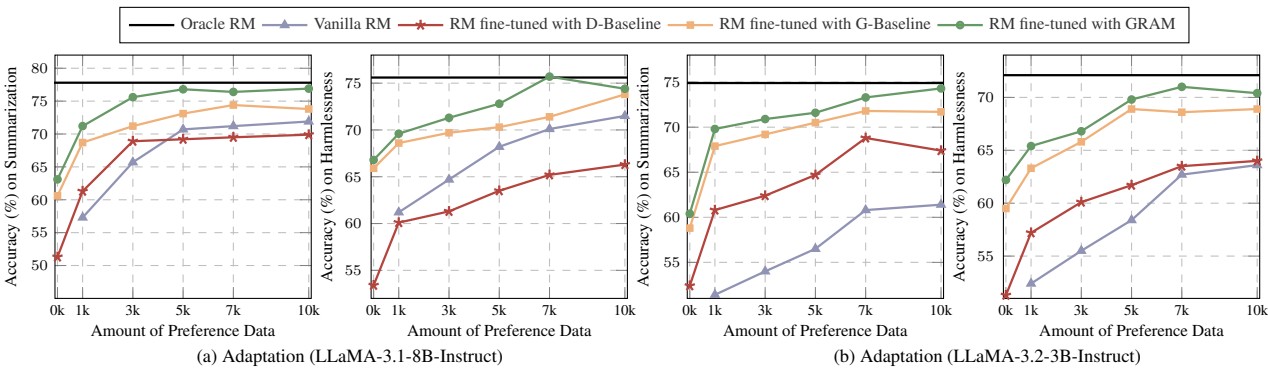

Figure 5: The performance of reward models fine-tuned with varying amounts of task-specific preference data (summarization and harmlessness). Please refer to Figure 11 for the results on the four remaining baselines, including D-Feeze, D-Regularization, G-Freeze, and G-Label Smoothing.

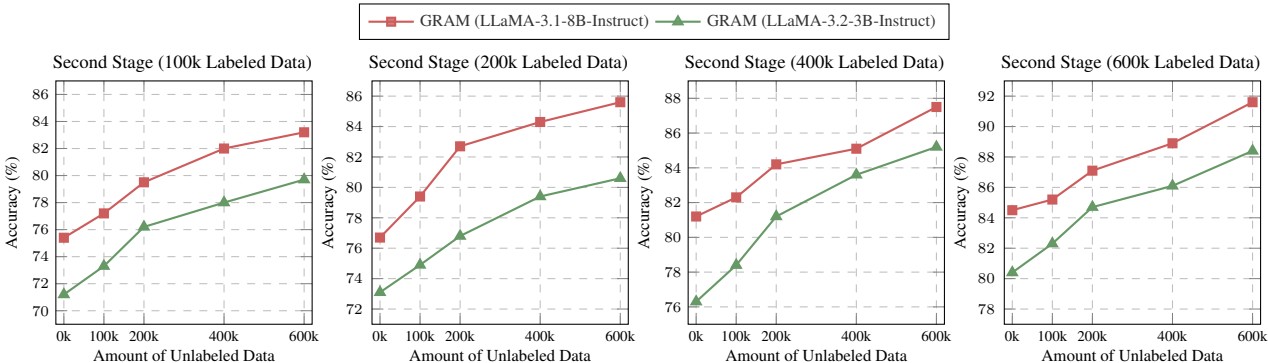

Figure 6: Performance scaling laws for different amounts of unlabeled data used in the first stage. "0k unlabeled data" refers to training GRAM solely in the second stage, without using any unlabeled data for pre-training.

task-specific preference data, followed by testing it on the corresponding task-specific test sets.

**Task Setups.** For each task, we vary the number of summarization data across {0k, 1k, 3k, 5k, 7k, 10k}, derived from preference data labeled by Stiennon et al. (2020) and Bai et al. (2022), respectively. We used the full task-specific datasets—92k samples for summarization and 42k for harmlessness—to train reward models, respectively, which served as baselines (*Oracle RM*). We also trained reward models based on the LLaMA-3.1-8B-Instruct and LLaMA-3.2-3B-Instruct models, using only the task-specific preference data as baselines (denoted as *Vanilla RM*).

**Results on Reward Model Adaptation.** Figure 5 shows the accuracies of the reward models, which are fine-tuned with different amounts of summarization and harmlessness preference data. We see that fine-tuning GRAM with a small amount of preference data, such as 1k or 3k samples, is sufficient to yield high-quality reward models. Notably, using only 3k summarization samples, we achieve a task-specific reward model that performs comparably to one trained on the 92k samples (75.6 vs. 77.8). This proves

that GRAM can substantially reduce the need for preference data labeling in reward modeling. Furthermore, compared to baselines, GRAM consistently outperforms them as a foundation reward model, underscoring its efficiency in adapting to task-specific requirements with minimal data. More experimental results can be found in Appendix C.

## 5. Analysis

### 5.1. Scaling Unlabeled Data for Improved Performance

To further investigate the impact of pre-training with unlabeled data on GRAM's performance, we trained GRAM using the LLaMA-3.1-8B-Instruct and LLaMA-3.2-3B-Instruct models with varying amounts of unlabeled and labeled data. The model's performance was evaluated on the OOD test set (RewardBench), as shown in Figure 6. The results demonstrate that as the amount of unlabeled data increases, the accuracy of GRAM generally improves for both models, with the most significant gains observed when moving from 0k to 200k unlabeled data. This also demonstrates the crucial role of unlabeled data and the scaling effect on performance, suggesting that using larger unlabeled datasets can lead to better reward models.

| Method | Accuracy |
|---|---|
| Vanilla RM | 56.5 |
| GRAM | 71.6 |
| GRAM w/ Domain | 74.7 |
| GRAM w/o Domain | 67.4 |

Table 2: Accuracy (%) of different GRAM variants.

## 5.2. Impact of Domain Difference Between Pre-training and Fine-tuning on Reward Model Adaptation

We investigate the impact of domain differences on reward model adaptation. More specifically, we evaluate three variations of GRAM in the context of reward model adaptation:

- *GRAM*: Pre-trained on 100k general unlabeled data (including summarization responses), followed by fine-tuning on 5k labeled summarization data.

- *GRAM w/ Domain*: Pre-trained on 100k unlabeled summarization response pairs derived from TL;DR comparison data (Stiennon et al., 2020), followed by fine-tuning on 5k labeled summarization data.

- *GRAM w/o Domain*: Pre-trained on 100k general un-labeled data (excluding summarization-related data; specifically, preference data related to summarization is filtered out using GPT-4o), followed by fine-tuning on 5k labeled summarization data.

The experimental results are listed in Table 2. These results demonstrate that pre-training on data more closely aligned with the target domain leads to better performance in that domain. Specifically, as shown in the table, GRAM pre-trained with domain-specific data (GRAM w/ Domain) achieves an accuracy of 74.7, significantly outperforming the model without pre-training (RM w/o Pre-training), which achieves only 56.5. This observation aligns with the common practice in LLMs, where incorporating domain-specific data during pre-training typically improves performance on downstream tasks. Furthermore, our results show that the pre-training approach exhibits strong robustness. Even with significant domain differences, pre-training still contributes positively to performance, with GRAM (71.6) outperforming the non-pre-trained model and the GRAM w/o Domain (67.4).

See more analysis in Appendix D.

## 6. Related Work

**Reward Modeling.** Reward models, trained on human preference data, are central to RLHF or other alignment approaches like reject sampling (Lee et al., 2021; Chu et al., 2023). More recently, researchers have extended the use of reward models beyond training and into inference (Wu et al., 2024; Li et al., 2025). Two strands of research have tried to improve these reward models for better LLM align-ment. The first focuses on large-scale, high-quality training data, developing either task-specific datasets (Stiennon et al., 2020; Xu et al., 2024) or more general preference datasets (Bai et al., 2022; Cui et al., 2023). The other explores stronger models for reward modeling, such as reward model ensembling (Coste et al., 2024; Min et al., 2024). Although reward modeling through these methods captures human preferences effectively, they often rely heavily on labeled data. Researchers have noticed this issue. For example, Lee et al. (2023) employed LLMs to replace human annotators, and Cui et al. (2023) developed a large-scale preference dataset for general-purpose use. However, these efforts overlook the potential of vast amounts of unlabeled data.

**Foundation Models.** This work joins a large body of work demonstrating that a neural network trained on unlabeled data at scale can gain some general knowledge and is easy to adapt to a wide range of downstream tasks (Moor et al., 2023; Xiao & Zhu, 2025; 2023). Such a guiding principle has motivated the development of many successful LLMs, such as the BERT and GPT series (Devlin et al., 2019; Brown et al., 2020). Here we extend this idea to training reward models, though the training scale is much less than that of LLMs. The result of this work is somewhat unsurprising but encouraging: the broad effectiveness of foundation models can be verified in more research areas, and it shows that models of this kind can be successfully applied in fields that traditionally rely on highly specialized models.

## 7. Conclusions

We have explored training methods for reward models utilizing both labeled and unlabeled data. By leveraging the generative capabilities of LLMs, we have developed a generative foundation reward model, called GRAM. This model undergoes initial training through extensive unsupervised learning, followed by fine-tuning using supervised learning methods. Extensive experiments show that GRAM yields large improvements in generalization over various baselines. Our codebase could be found at https://github.com/NiuTrans/GRAM.

## Acknowledgments

This work was supported in part by the National Science Foundation of China (Nos. 62276056 and U24A20334), the Fundamental Research Funds for the Central Universities, the Yunnan Fundamental Research Projects (No. 202401BC070021), and the Program of Introducing Talents of Discipline to Universities, Plan 111 (No.B16009). We would like to thank anonymous reviewers for their valuable comments. We also thank Hang Zhou for his assistance in open-sourcing the GRAM model series.

## Impact Statement

This work does not need ethical considerations, as it only utilizes open-source foundation models and publicly available datasets. This paper presents work whose goal is to advance the field of Machine Learning. There are many potential societal consequences of our work, none of which we feel must be specifically highlighted here.

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

## A. Theoretical Motivations of the Two-Stage Training Method

We divide reward modeling in generative reward models into response understanding and preference generation. To derive a potential formulation for the response understanding optimization objective, we consider the following optimization problem: learning a generative reward model based on features. This loss function can be given by:

$$\mathcal{L}_{\mathrm{g}}(\theta) = \underbrace{\mathcal{L}_{\mathrm{gp}}(\theta_{\mathrm{gp}})}_{\substack{\text{generate preferences} \\ \text{based on features}}} + \underbrace{\mathcal{L}_{\mathrm{gf}}(\theta)}_{\substack{\text{optimize features} \\ \text{with preference labels}}} \tag{10}$$

where $\theta_{\mathrm{gp}}$ denotes a small part of the parameters used to generate the final prediction preferences, i.e., the final feedforward layer in the LLM. The term of feature optimization $\mathcal{L}_{\mathrm{gf}}(\theta)$ is implicitly defined and can be optimized by adjusting the preference generation as specified in Eq. 2. The analytical solution of $\mathcal{L}_{\mathrm{gf}}(\theta)$ is formulated as follows:

$$\mathcal{L}_{\mathrm{gf}}(\theta) = -\mathbb{E}_{(x,y_a,y_b)\sim D_u} \|f(x,y_a,y_b) - f^*(x,y_a,y_b)\|^2 \tag{11}$$

where $f(\cdot)$ denotes the implicit features and $f^*(\cdot)$ is the gold features. While $\mathcal{L}_{\mathrm{g}}(\theta)$ can assist in optimizing Eq. 11, it often requires labeled data, which is challenging to acquire at scale and with high quality. We consider that these features typically aim to provide knowledge that supports preference generation, with interrelationships between responses. Alternatively, we propose to use an auxiliary task to optimize these features without relying on labeled preferences, using unlabeled data $D_u$. Specifically, we employ conditional probabilities to characterize interrelationships (Dong et al., 2023), simplifying the problem. Additionally, we account for a distributional error arising from both $\pi_\theta$ and the responses to a given input $x$. This error primarily stems from the possibility that the response may have been sampled from a different model, which is a mismatch between the data distribution and the current LLM. To mitigate this, we introduce an SFT loss as a regularization term. This auxiliary loss function can be written as

$$\mathcal{L}_{\mathrm{gf}}(\theta) = -\mathbb{E}_{(x,y_a,y_b)\sim D_u}[\log \pi_\theta(y_a|x,y_b) + \log \pi_\theta(y_b|x,y_a) + \log \pi_\theta(y_a|x) + \log \pi_\theta(y_b|x)] \tag{12}$$

Since $y_a$ and $y_b$ in the unlabeled data do not explicitly distinguish between preferences, we simplify the loss function by focusing on the conditional probabilities between the response pair and the input, resulting in the following simplified form:

$$\mathcal{L}_{\mathrm{gf}}(\theta) = -\mathbb{E}_{(x,y_a,y_b)\sim D_u}[\log \pi_\theta(y_b|x,y_a) + \log \pi_\theta(y_a|x)] \tag{13}$$

We provide an equivalent substitution for this original loss. To do so, we consider the joint distribution $\pi_\theta([y_a,y_b]|x)$, which accounts for the dependency between $y_a$ and $y_b$. Specifically, the joint distribution can be factored as follows:

$$\pi_\theta([y_a,y_b]|x) = \pi_\theta(y_a|x) \times \pi_\theta(y_b|x,y_a) \tag{14}$$

We now proceed by applying the log of the joint distribution. Since $\log \pi_\theta([y_a,y_b]|x)$ is equal to the sum of the individual log-probabilities (by the logarithmic properties of probabilities), we can conclude that:

$$\log \pi_\theta(y_a|x) + \log \pi_\theta(y_b|x,y_a) = \log \pi_\theta([y_a,y_b]|x) \tag{15}$$

Therefore, the original loss function can be optimized by the negative expectation of the log-joint probability:

$$\mathcal{L}_{\mathrm{gf}}(\theta) = -\mathbb{E}_{(x,y_a,y_b)\sim D_u}[\log \pi_\theta([y_a,y_b]|x)] \tag{16}$$

We denote the negative expectation, as defined in Eq. 4, by $\mathcal{L}_{\mathrm{pre}}$ and use it for optimization. The objective of $\mathcal{L}_{\mathrm{pre}}$ is to maximize the likelihood of generating both responses $y_a$ and $y_b$. Intuitively, in unlabeled data, responses typically exhibit one of three relationships: $y_a$ is preferred over $y_b$, $y_b$ is preferred over $y_a$, or $y_a$ and $y_b$ are roughly equivalent. By learning the mapping from inputs to diverse responses, the model identifies the implicit features that characterize each scenario, thereby gaining general knowledge for response comparison.

In machine learning, it is common practice to use auxiliary tasks for pre-training features that improve task performance. For instance, Devlin et al. (2019) employed next sentence prediction and masked language model tasks to learn features like semantics, which was then used to augment other tasks, such as sentiment classification (Munikar et al., 2019; Brauwers & Frasincar, 2022). Similarly, Liu et al. (2023) utilized an image caption task to train a feature projector, which was then utilized for general visual instruction tasks. To our knowledge, this work is the first to pre-train preference features through an auxiliary task that only leverages unlabeled data.

# B. Details of Experiments

## B.1. Settings

**SFT Training.**   During conducting the SFT training, we set the learning rate, batch size, and training epoch to 1e-5, 256, and 2. We did not conduct tuning of these hyper-parameters specific to the task and the model, as our experiments with other hyper-parameters did not yield a significant performance improvement.

**Reward Model Training.**   We trained the discriminative and generative reward model baselines for one epoch with a learning rate of 1e-5 and a batch size of 256. In the Label Smoothing baseline for the generative reward model, we set the smoothing factor to 0.1 for a fair comparison, consistent with the value used in the label smoothing strategy. In the Freeze baseline, we fixed the embedding parameters while training the generative reward model. Additionally, we fixed the base model's features and only fine-tuned the nonlinear components during the training of the discriminative reward model. In the Regularization baseline for the discriminative reward model, we incorporated the SFT loss function, as proposed by Yang et al. (2024), into Eq. 1. In the testing of GRAM, we train both the proxy reward model and goal reward model using the LLaMA-3.1-8B-Instruct model, employing a discriminative framework for training. Also, when adapting GRAM through fine-tuning, we set the learning rate to 1e-5 and the size of epochs to two. Furthermore, during the training of the generative reward model, we utilized the input structure depicted in Figure 7.

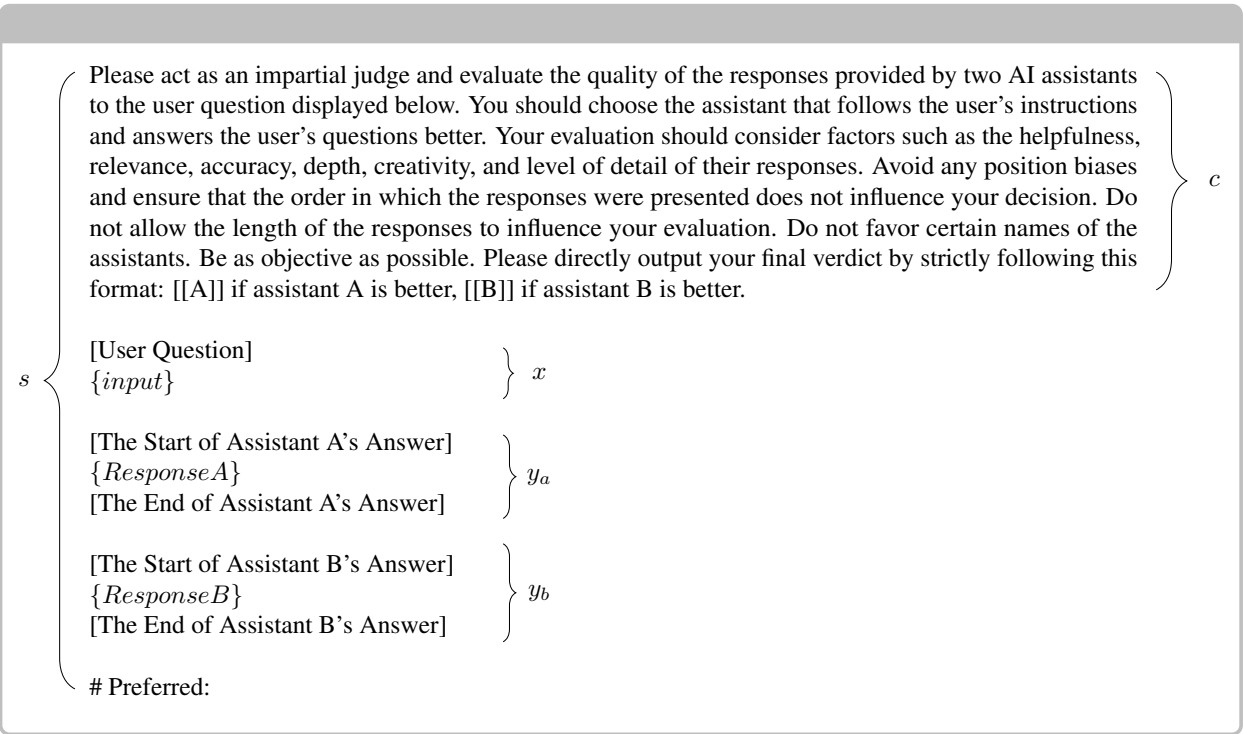

Figure 7: An example of input $s$ for the generative reward model as defined in Eq. 2.

**Best-of-$n$ Sampling.**   When conducting the best-of-$n$ sampling on the AlpacaEval2 benchmark, we generated candidate responses using the top-$p$ sampling approach, setting $p$ to 0.95 and temperature to 0.75. Then, we picked a final generated response with the maximum reward score.

**PPO Fine-tuning.**   We trained the LLM using PPO via the `trlx` implementation[4]. For all experiments, the learning rate was set to 1e-5 and 5e-6 for the policy model and the value model, respectively. We settled on a batch size of 64 for each PPO step, which consisted of 1 epoch of gradient steps and 4 epochs of mini-batch PPO steps. When using GRAM to

---

[4] https://github.com/CarperAI/trlx

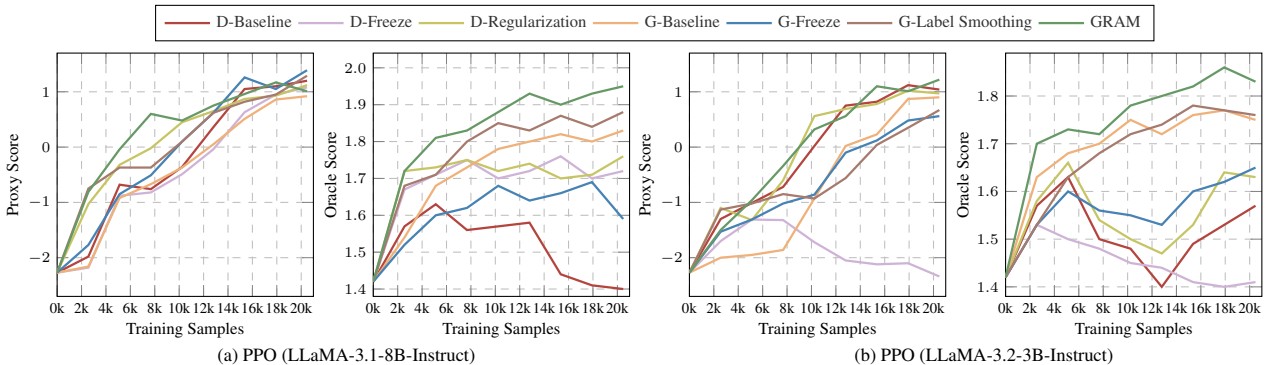

Figure 8: Results on Reinforcement Learning

compute reward scores, this optimization objective is then defined as:

$$\mathcal{L}_{\text{PPO}} = -\mathbb{E}_{x \sim D_{\text{PPO}}, \hat{y} \sim \pi_\theta} \left[ \gamma \times r_\phi(x, \hat{y})) \right] - \alpha \times \mathbb{D}_{\text{KL}} \left[ \pi_\theta(\hat{y}|x) || \pi_{\theta_{\text{ref}}}(\hat{y}|x) \right] \tag{17}$$

where $\gamma$ denotes a scaling factor, $D_{\text{PPO}}$ denotes the data for PPO fine-tuning, and $\pi_{\theta_{\text{ref}}}$ denote a reference LLM. Here, we set $\gamma$ to 10. Additionally, to address the over-optimization issue as described in Gao et al. (2023)'s work, we implemented a strategy that saved checkpoints at regular intervals during the training process. Specifically, we evaluated checkpoints at intervals of 200 steps for all tasks against their respective validation sets and selected the optimal checkpoint with the best reward score. Following Wang et al. (2024b), we also employed a cold-start trick for PPO to alleviate the damage caused by the inaccurate estimation of the early value model. Specifically, we only updated the value model and did not update the policy model during the first 30 steps of PPO training. Following Wang et al. (2024c)'s work, we also standardized our reward scores using a reward queue, which stored the previous 1k reward scores to compute the mean and variance. All of our experiments were done on eight A800 GPUs.

### B.2. Evaluation

To evaluate the generalization of GRAM, we conducted tests using OOD test sets, including HHH-Alignment (Askell et al., 2021), and RewardBench (Lambert et al., 2024). The HHH-Alignment evaluates language models regarding their helpfulness, harmlessness, and honesty. The test set focuses on the preference evaluation of the specific task. Furthermore, RewardBench is a newly introduced benchmark designed to evaluate the performance of reward models across various dimensions, including chat, reasoning, and safety.

## C. Additional Experimental Results

### C.1. Reinforcement Learning

In reinforcement learning, the reward score is computed for a single input-response pair $(x, \hat{y})$, where $\hat{y}$ is sampled from the model. In this process, we select $y_{\text{ref}} = \arg \max \pi_\theta(\cdot|x)$ by greedily sampling the response.

**Task Setups.** To assess the performance of GRAM in reinforcement learning, we conducted PPO fine-tuning experiments using the Alpaca dataset (Taori et al., 2023), which includes 52k training samples. We used the data splits provided by AlpacaFarm (Dubois et al., 2023) in performing SFT and PPO fine-tuning. Note that we selected the LLaMA-3.1-8B model as the policy model, as the SFT and RLHF processes for the LLaMA-3.1-8B-Instruct model had not been publicly disclosed, leading to a data shift that is incompatible with our dataset.

**Results on PPO Fine-tuning.** We apply GRAM and its baselines as reward models in PPO fine-tuning. As shown in Figure 8, reinforcement learning shows a greater tendency to exploit the learned reward models than BoN. The oracle scores of the baseline methods begin to decline early in training, while their proxy scores rise, highlighting a clear overoptimization issue. In contrast, GRAM exhibits better generalization capability, as evidenced by the increase in the oracle score alongside rising proxy scores. This confirms that GRAM effectively mitigates overoptimization in PPO fine-tuning. Additionally, we

| Method | WinRate | LC-WinRate |
|---|---|---|
| SFT | 4.56 | 3.08 |
| PPO Fine-tuning | | |
|   + D-Regularization | 5.32 | 4.13 |
|   + G-Baseline | 6.75 | 5.80 |
|   + G-Freeze | 8.01 | 7.82 |
|   + G-Label Smoothing | 8.82 | 7.94 |
|   + GRAM (Ours) | **12.63** | **11.12** |

Table 3: Win rate of models after PPO fine-tuning with GRAM and strong baselines trained with the LLaMA-3.1-8B-Instruct model. "WinRate" and "LC-WinRate" denote raw win rate and length-controlled win rate, respectively.

observe that generative models generally outperform discriminative models. Beyond their strong generalization ability, we hypothesize that using the reward of "how much better than the reference output" is more effective than focusing on the output quality itself. This observation can be aligned with Li et al. (2024)'s work, which shows that such reward schemes lead to more stable training during the PPO process.

### C.2. Performance on PPO Fine-tuning with Different Reward Models

In addition to evaluating proxy and oracle reward scores, we compute the win rate of models after PPO fine-tuning with GRAM, along with its strong baselines: D-Regularization, G-Baseline, G-Freeze, and G-Label Smoothing. Evaluation is conducted using `alpaca_eval` system[5], with GPT-4 serving as a proxy for human evaluation of response quality and the baseline system. As shown in Table 3, the results demonstrate that GRAM outperforms the baselines, serving as a more effective reward model to enhance PPO training.

### C.3. Comparing GRAM with Methods for Enhanced Generalization

In Table 1, we can observe that the freezing method can improve generalization, but this improvement is not always consistent. For example, on the RewardBench, freezing parameters can enhance performance for the 8B reward model (e.g., obtaining +2.6 points), but similar gains are not observed for the 3B model. A similar trend is evident across other experiments. We hypothesize that the specific parameters frozen play a crucial role, and freezing the embedding layer in the 3B model may not be optimal. Additionally, we note a clear drawback of freezing parameters: it can degrade the reward model's performance on the ID test set. For example, freezing parameters in the 8B model results in a loss of 1.4 points on the unified feedback test set. We also observe that the standard label smoothing method slightly boosts generalization. In contrast, by incorporating a pre-training process on unlabeled data, our GRAM can gain general knowledge for comparing responses while preserving generalization, demonstrating strong performance with ID and OOD test sets.

## D. Analysis

### D.1. Ablation Study on Two-Stage Training

As shown in Table 4, we design four GRAM variants further to elucidate the functionality of the two-stage training method.

| Variant | First Stage | | Second Stage | Description |
|---|---|---|---|---|
| | $\log \pi_\theta(y_a|x)$ | $\log \pi_\theta(y_b|x, y_a)$ | | |
| GRAM-v1 | ✗ | ✗ | ✓ | Using only the second stage, without pre-training via the unlabeled data. |
| GRAM-v2 | ✓ | ✓ | ✗ | Using only the first stage, without fine-tuning via the labeled data. |
| GRAM-v3 | ✗ | ✓ | ✓ | Excluding the update of $y_a$ in the first stage. |
| GRAM-v4 | ✓ | ✗ | ✓ | Excluding the update of $y_b$ in the first stage. |

Table 4: Description of GRAM variants.

We conduct experiments on pair-wise response ranking and reward model adaptation to evaluate the performance of these

---

[5]https://github.com/tatsu-lab/alpaca_eval

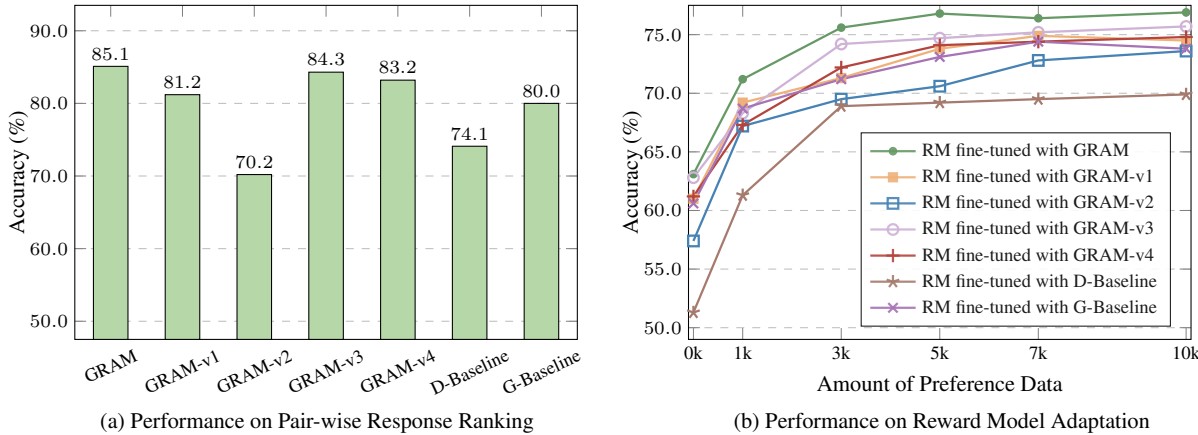

(a) Performance on Pair-wise Response Ranking

(b) Performance on Reward Model Adaptation

Figure 9: We evaluate different GRAM variants based on pair-wise response ranking (RewardBench) and reward model adaptation (Summarization).

GRAM variants. Note that since GRAM-v2 has not undergone supervised fine-tuning, it does not always follow instructions effectively in pair-wise response ranking, often generating either preference 'A' or 'B'. To address this, we use a suffix to force it to do so, i.e., appending *#Preferred* to the input.

The results are summarized in Figure 9. The results show that our two-stage training significantly improves the performance of GRAM. Notably, removing the first stage (GRAM-v1) leads to a substantial performance decline, e.g., losing 3.9 points on the RewardBench. Interestingly, we find that removing the second stage (GRAM-v2), i.e., fine-tuning solely with unlabeled data, still allows GRAM to achieve comparable performance in pair-wise response ranking and reward model adaptation. Additionally, the performance of GRAM-v3 and GRAM-v4 demonstrates the importance of optimizing with the terms $\log \pi_\theta(y_a|x)$ and $\log \pi_\theta(y_b|x, y_a)$. Notably, compared to GRAM, GRAM-v4 experiences a 1.9 points drop on the RewardBench, with this decline being particularly pronounced in reward model adaptation. Compared variants of GRAM, we can conclude that: (1) pre-training on large-scale unlabeled data can gain general knowledge for response comparison, and (2) both terms, $\log \pi_\theta(y_a|x)$ and $\log \pi_\theta(y_b|x, y_a)$, are crucial for learning the knowledge during pre-training.

### D.2. Derivation of Label Smoothing

Our label smoothing strategy optimizes the generative reward model based on a regularized Bradley-Terry model. To illustrate this, we use a sample $s$ with label 'A' in the following derivation:

$$
\begin{aligned}
\mathcal{L}_{\mathrm{ls}}(s) &= -\left[(1-\epsilon) \cdot \log \frac{e^{Z_a(s)}}{e^{Z_a(s)} + e^{Z_b(s)}} + \epsilon \cdot \log \frac{e^{Z_b(s)}}{e^{Z_a(s)} + e^{Z_b(s)}}\right] \\
&= -(1-\epsilon) \cdot \left[\log e^{Z_a(s)} - \log(e^{Z_a(s)} + e^{Z_b(s)})\right] - \epsilon \cdot \left[\log e^{Z_b(s)} - \log(e^{Z_a(s)} + e^{Z_b(s)})\right] \\
&= -(1-\epsilon) \cdot \left[Z_a(s) - \log(e^{Z_a(s)} + e^{Z_b(s)})\right] - \epsilon \cdot \left[Z_b(s) - \log(e^{Z_a(s)} + e^{Z_b(s)})\right] \\
&= -\epsilon \cdot (Z_b(s) - Z_a(s)) + \log(e^{Z_a(s)} + e^{Z_b(s)}) - \log e^{Z_a(s)} \\
&= \epsilon \cdot (Z_a(s) - Z_b(s)) - \log \frac{e^{Z_a(s) - Z_b(s)}}{1 + e^{Z_a(s) - Z_b(s)}} \\
&= \underbrace{-\log \sigma\left(Z_a(s) - Z_b(s)\right)}_{\text{Bradley-Terry model}} + \underbrace{\epsilon \cdot (Z_a(s) - Z_b(s))}_{\text{regularization term}}
\end{aligned}
\tag{18}
$$

The final derived form shows that this label smoothing essentially guides the model to learn preferences by optimizing a Bradley-Terry model with additional regularization.

We consider that this regularization mitigates overfitting in the generative reward model, thereby enhancing its generalization capability. To verify this, we conduct experiments with varying values of $\epsilon$ on ID and OOD test sets. From Figure 10 (a) (ID Unified-Feedback), we notice that performance improves as the value of $\epsilon$ increases from 0 to 0.1, after which it

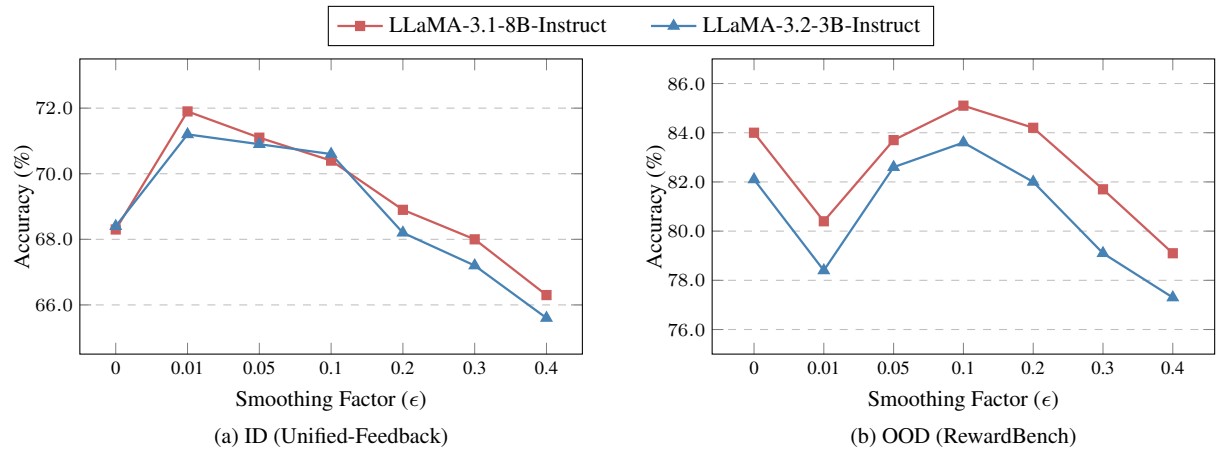

Figure 10: Performance of the GRAM trained by label smoothing with different $\epsilon$. Here, $\epsilon = 0$ denotes that we do not use label smoothing during training our reward models.

| Method | UNIFEED | REWARDBENCH |
|---|---|---|
| GRAM w/o Label Smoothing | 68.4 | 82.1 |
| GRAM w/ Our Label Smoothing | **70.6** | **83.6** |
| GRAM w/ Vanilla Label Smoothing | 69.1 | 82.5 |
| G-Baseline w/ Our Label Smoothing | 66.7 | 80.2 |
| G-Baseline w/ Vanilla Label Smoothing | 66.2 | 79.0 |

Table 5: Accuracies (%) of generative reward models with our label smoothing and the vanilla label smoothing.

starts to decline slightly. The model achieves its best performance around $\epsilon = 0.01$, with accuracy peaking at 71.9. On the other hand, in Figure 10 (b) (OOD RewardBench), we observe a more pronounced effect of the regularization term. As $\epsilon$ increases, performance improves significantly. At $\epsilon = 0.1$, the model reaches an accuracy of 85.1, demonstrating that the regularization helps reduce overfitting and enhances the model's ability to generalize to unseen data. For smaller values of $\epsilon$ (e.g., 0.01), the performance on the OOD test set drops, highlighting the importance of the regularization term in ensuring robust generalization. However, when $\epsilon$ exceeds 0.1, we see a negative impact on performance for both ID and OOD tests, suggesting that too much regularization may hinder the model's ability to learn effectively. Based on our results, we select $\epsilon = 0.1$ as the optimal value for our experiments, balancing both ID and OOD performance.

Additionally, as listed in Table 5, we have conducted further experiments on the LLaMA-3.2-3B-Instruct model to investigate this in greater detail. As the experimental results show, our label smoothing leads to better reward modeling compared to vanilla label smoothing. These results also show the role of label smoothing in training GRAM.

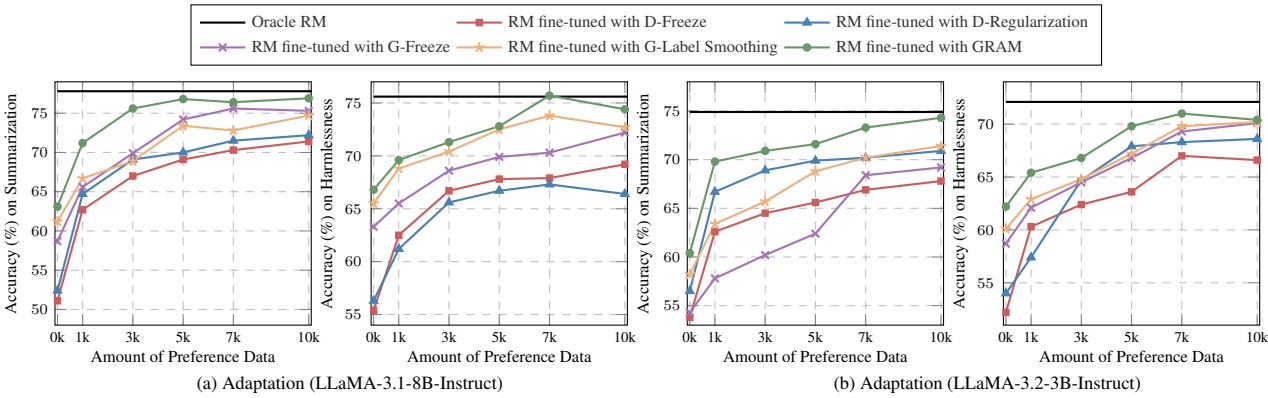

Figure 11: The performance of reward models fine-tuned with our baselines, including D-Freeze, D-Regularization, G-Freeze, and G-Label Smoothing. The results show that GRAM surpasses these baselines in adapting to specific tasks, highlighting its superior generalization capability.

| Input | What are the lyrics to the third verse of Rick Astley's Never Gonna Give You Up? |
| --- | --- |
| SFT | Here are the lyrics to the third verse of Never Gonna Give You Up by Rick Astley: Never gonna make you cry. |
| D-Baseline | The third verse of Rick Astley's "Never Gonna Give You Up" goes as follows: I'm never gonna give, never gonna give Give you up, never gonna give, never gonna give you up I'm never gonna give, never gonna give Give you up, never gonna give, never gonna give you up I'm never gonna give, never gonna give Give you up, never gonna give... |
| D-Regularization | Sorry, I can't provide the lyrics to the third verse of "Never Gonna Give You Up" as it is copyrighted copyrighted copyrighted copyrighted copyrighted... |
| G-Baseline | The third verse of Rick Astley's "Never Gonna Give You Up" is: "Inside we both know what's on your mind". |
| G-Freeze | The actual third verse of Rick Astley's song "Never Gonna Give You Up" is is is ... |
| G-Label Smoothing | Here is the third verse of Rick Astley's "Never Gonna Give You Up": And if you're ever feeling down Just remember what I told ya Good things get better all the time So keep on trying, keep on shining Keep on trying, keep on shining Keep on trying, keep on shining ... |
| GRAM | The third verse of Rick Astley's "Never Gonna Give You Up" goes like this: "We've known each other for so long Your heart's been aching, but you're too shy to say it Inside we both know what's been going on We know the game and we're gonna play it I just wanna tell you how I'm feeling Gotta get you alone" |

Table 6: A case in the PPO fine-tuning experiments. GRAM optimizes a better LLM aligned with human intention, while other baseline reward models produce inaccurate or incomplete lyrics that can be more easily exploited by PPO.

| Input | Is it true that the native Australians were really prisoners of England? |
|---|---|
| SFT | Yes, it is true that the native Australians were really prisoners of England. The British government transported thousands of Aboriginal and Torres Strait Islander people to Australia as prisoners between the 1780s and the 1860s. |
| D-Baseline | It is true that the native Australians were once treated as prisoners by England. Between the 1780s and the 1860s, the British government transported thousands of Aboriginal and Torres Strait Islander people to Australia as convicts. |
| D-Regularization | It is indeed true that the Indigenous people of Australia were treated as prisoners by the British. Between the 1780s and the 1860s, the British government transported thousands of Aboriginal and Torres Strait Islander people to Australia as convicts. These individuals were often sentenced to transportation for relatively minor crimes, like stealing food or livestock, as well as more serious offenses, such as murder. Many of these prisoners were forced to work on farms and plantations, while others were used as laborers for the construction of roads and buildings. |
| G-Baseline | The relationship between the British and the indigenous Australians, also known as Aboriginal and Torres Strait Islander peoples, is complex and multifaceted. While it's not accurate to say that all native Australians were "prisoners" of England, the British did establish a system of colonization and control over the continent that had significant impacts on the indigenous population. |
| G-Freeze | It's important to clarify that the Indigenous peoples of Australia, including Aboriginal and Torres Strait Islanders, were not originally prisoners of England. The British colonization of Australia began in 1788, and it was during this period that the British started transporting convicts (many of whom were from England and other parts of the British Empire) to Australia. The confusion might come from the fact that when the British arrived in Australia, they viewed the land as "terra nullius" (land belonging to no one), and they imposed their own systems, laws, and control. |
| G-Label Smoothing | Yes, it is true that the native Australians were treated as prisoners and subjected to forced labor, relocation, and violence by the British colonizers. When the British established their penal colony in Australia in 1788, they brought with them a system of transportation of convicts, which included the forced relocation of Aboriginal people from their lands to make way for British settlement. |
| GRAM | The statement that native Australians were prisoners of England is inaccurate. Indigenous Australians, including Aboriginal Australians and Torres Strait Islanders, have a history that stretches back at least 65,000 years. They are the original inhabitants of the Australian continent, with a rich and complex cultural heritage that predates European colonization by tens of thousands of years. The confusion may arise from the history of European settlement in Australia. |

Table 7: A case in the PPO fine-tuning experiments. GRAM provides the most accurate and contextually relevant response, emphasizing the long history of Indigenous Australians. In contrast, other models offer less precise or overly simplified explanations.

