# OpenReview forum: "GRAM: A Generative Foundation Reward Model for Reward Generalization"
_ICML.cc/2025/Conference — ICML 2025 poster_

### Official Review · Reviewer_Jdyw · 2025-02-26

**Overall Recommendation:** 3

**Summary:**

The paper proposes a method for training a generative reward model that generalizes better across domains with minimal fine-tuning. The approach involves a two-step training process: large-scale unsupervised learning followed by supervised fine-tuning on labeled data. The paper also demonstrates that applying label smoothing during training can be interpreted as optimizing a regularized pairwise ranking loss, highlighting a connection between training discriminative and generative reward models. The resulting 'foundation reward model' outperforms discriminative baselines on tasks including response ranking, best-of-n sampling, and task adaptation with fine-tuning.

**Claims And Evidence:**

The main claims about the proposed GRAM are that it generalizes better than both discriminative and prior generative reward models, outperforming them on response ranking, reinforcement learning from human feedback, and task adaptation with fine-tuning.

Figure 2 and Table 1 present the main results supporting the generalization claim, showing that GRAM performs better on out-of-distribution test data, including RewardBench and HHH-Alignment. However, the results in Figure 2 and for Llama-3.1-8B-Instruct in Table 1, where GRAM underperforms compared to discriminative reward models on in-distribution test data, raise the question of whether this is due to discriminative models overfitting to in-distribution data rather than indicating a better generalization capability. The results for Llama-3.2-3B-Instruct in Table 1 are arguably more convincing, as GRAM outperforms on both in-distribution and out-of-distribution test data.

The best-of-n sampling results also show that GRAM is less susceptible to overoptimization, which appears closely related to its generalization performance. However, the paper does not include reinforcement learning experiments where an LLM is trained to optimize the reward model using RL or methods like DPO that optimize for the same objective. Without reinforcement learning results, the claim in the abstract that GRAM outperforms in reinforcement learning from human feedback seems less convincing.

**Essential References Not Discussed:**

Not that I am aware of.

**Experimental Designs Or Analyses:**

Overall, the use of two open instruct models for reward model training, along with common reward model benchmarks such as RewardBench and HHH-Alignment, for comparison with both discriminative reward models and strong LLMs like GPT-4, seems sound. More ablations, such as evaluating how much the proposed 'pre-training' helps, would have been insightful.

**Methods And Evaluation Criteria:**

The proposed 'pre-training' step, where a generative reward model is trained to generate two responses per input, makes some sense. However, the extent to which this training improves performance beyond fine-tuning is not thoroughly demonstrated empirically. Additionally, the paper does not discuss in detail how the two responses should be prepared -- specifically, how to prompt an LLM to generate multiple responses (e.g., simply different responses or responses of varying quality). Also, some discussion on how the proposed training scheme could be extended to multi-objective settings, where two responses are evaluated according to multiple criteria, would have been more insightful.

The evaluation on several common use cases of reward models, including response ranking, alignment, and reward model adaptation, seems appropriate for the problem. To strengthen the results, RL fine-tuning using the different reward models would have been beneficial.ㅒ

**Other Comments Or Suggestions:**

Typos:
- "an" $\rightarrow$ "a" near line 104.
- "Best-of-n" $\rightarrow$ "best-of-n" near line 146.
- "Opeen" $\rightarrow$ "Open" near line 282.

**Other Strengths And Weaknesses:**

None.

**Questions For Authors:**

Q. Have you considered using reasoning models for generative reward models?

**Relation To Broader Scientific Literature:**

Reward models are essential for evaluating and aligning generative models, and, as the paper also suggests, much less work has been done on developing 'foundation' reward models compared to foundation generative models. This paper contributes to the recent efforts in developing generative models, particularly those considered 'foundational,' where pre-training is used to train domain-agnostic models that can be easily adapted to different domains with minimal extra data. More efforts will need to be devoted to developing such generic reward models in an unsupervised manner for continued improvement of generative models going forward.

**Theoretical Claims:**

No major theoretical claims are made in the paper.

---

> ### Author Rebuttal · Authors · 2025-03-31
>
> Dear Reviewer Jdyw,
>
> We appreciate that you find "the superior generalization of our GRAM compared to both discriminative and prior generative reward models" and our pre-training approach is "make sense".
>
> We will provide explanations for the main points that you are concerned about.
>
> ---
> >*W1: The results in Figure 2 and Table 1, where GRAM underperforms compared to discriminative models on ID data, raise the question of whether this reflects overfitting by discriminative models rather than better generalization.*
>
> We would like to clarify this question in two aspects.
> - The ID test set evaluates the ability of reward modeling to learn human preferences from labeled data. Our goal is to excel in reward modeling and generalization, i.e., obtaining strong performance on ID and OOD tasks simultaneously. Thus, the LLaMA-3.1-8B-Instruct results in Table 1 show our method's effectiveness, achieving the best OOD results and second-best ID results.
> - Although GRAM underperforms compared to the ```Discriminative RM+Regularization``` when using LLaMA-3.1-8B-Instruct, it significantly outperforms both the ```Discriminative RM (Baseline)``` and ```Discriminative RM+Freeze```, supporting the validity of our method. We also see that ```Regularization``` may not be a universally effective method across all models, e.g., on the LLaMA-3.2-3B-Instruct model, ```Regularization``` performs notably worse than GRAM, suggesting that its effectiveness could be model-dependent.
>
> The partial results of Table 1 are shown below:
>
> |Method|UniFeed(ID)|RewardBench(OOD)|
> |:-|:-:|:-:|
> |**LLaMA-3.1-Instruct**|
> |Discriminative RM (Baseline)|69.3|74.1|
> |Discriminative RM+Freeze|66.6|74.9|
> |Discriminative RM+Regularization|**72.7**|77.4|
> |GRAM (Ours)|70.4|**85.1**|
> |**LLaMA-3.2-Instruct**|
> |Discriminative RM (Baseline)|68.3|72.8|
> |Discriminative RM+Freeze|63.0|70.5|
> |Discriminative RM+Regularization|65.6|71.2|
> |GRAM (Ours)|**70.6**|**83.6**|
>
> >*W2: The paper lacks reinforcement learning experiments where an LLM optimizes the reward model using RL.*
>
> We apologize for any misunderstanding regarding the RL experiments. In fact, we have already included reinforcement learning experiments in Appendix C.1. Specifically, we comprehensively compare PPO performance with different reward models, as shown in Figure 8 and Table 2. The results align with the response ranking, showing that GRAM exhibits strong generalization and provides more accurate rewards. To respond to your concern, and in the revised version, we promise to include partial results in the body rather than only in the appendix.
>
> >*Q1: The paper does not detail how to prepare the two responses, particularly how to prompt an LLM for multiple responses (e.g., simply different responses or responses of varying quality).*
>
> Thanks for this insightful suggestion! We have explored this issue with two GRAM variants.
> - GRAM-Sim-Diff: We explore the impact of response diversity on pre-training performance. Specifically, we select the 200k response pairs with the highest semantic differences from a dataset of 600k responses and compare them to 200k randomly selected pairs.
> - GRAM-Qua-Diff: We test quality differences by scoring responses with GPT-4o on a 0-5 scale and compare the 200k pairs with the largest quality differences to 200k random pairs.
>
> The results are as follows:
> |Method|RewardBench|
> |:-|:-:|
> |GRAM-Sim-Diff|74.8|
> |GRAM-Qua-Diff|75.4|
> |GRAM|**76.2**|
>
> We find that randomly selected pairs outperformed the carefully selected ones, possibly due to increased diversity from randomness. We also see that larger quality differences have little impact on reward model training, with random selection performing better. This is also shown in previous work (Filtered Direct Preference Optimization), where larger quality differences benefited DPO training but not reward model training. We promise to add more experiments and analysis in the revised version.
>
> >*Q2: A discussion on extending the training scheme to multi-objective settings, where responses are evaluated by multiple criteria, would be insightful. Have you considered using reasoning models for generative reward models?*
>
> Thanks for your valuable suggestion! GRAM can easily be extended to multi-objective or reasoning settings, and this extension can be implemented in stage two. Specifically, we would describe the objectives, e.g., fluency and accuracy, in the prompt $c$. When generating preferences, we can use the format ```#Fluency Preferred: [labeled token] #Accuracy Preferred: [labeled token]``` instead of ```#Preferred: [labeled token]```. Similarly, we can also add the CoT in front of the preferences. We can also use a reasoning model to develop GRAM further. These improvements don’t require changes to our pre-training, highlighting its robustness and scalability for reward modeling. We promise to include more analysis in the revised version.
>
> ---
> We truly appreciate your positive feedback on our paper!
>
> Best,
>
> Authors

---

### Official Review · Reviewer_pUcN · 2025-03-09

**Overall Recommendation:** 3

**Summary:**

This paper proposes an interesting reward model training method using both unlabeled and labeled data. Building on the generative models in LLMs, the authors develop a generative reward model that is first trained via large-scale unsupervised learning and then fine-tuned via supervised learning. This method produces a foundation reward model, which can be applied to different tasks with little or no further fine-tuning effort, including response ranking, reinforcement learning from human feedback, and task adaptation with fine-tuning, achieving performance improvements over baseline models.

**Claims And Evidence:**

yes

**Essential References Not Discussed:**

No

**Experimental Designs Or Analyses:**

yes

**Methods And Evaluation Criteria:**

yes

**Other Comments Or Suggestions:**

The authors should conduct more evaluations and provide more details/discussions as the weakness part comment.

**Other Strengths And Weaknesses:**

**Strengths**:

1. The proposed training paridigm for reward model is interesting and sound.

2. The proposed method is extensively evaluated across different tasks.

**Weaknesses**:

1. I am wondering whether using such enhanced RM could improve the reasoning ability of existing LLMs. More downstream evaluations should be conducted.

2. The paper does not introduce much details about "large-scale unsupervised learning", the influence of pretraining data on target domain should also be discussed.

**Questions For Authors:**

Would the domain difference between pretraining and finetuning has a big impact on RM performance?

**Relation To Broader Scientific Literature:**

No

**Theoretical Claims:**

yes

---

> ### Author Rebuttal · Authors · 2025-03-31
>
> Dear Reviewer pUcN,
>
> We sincerely thank the reviewer for your positive and insightful feedback.
> We greatly appreciate your recognition of our paper's proposed training paradigm for the reward model as "interesting and sound", and are pleased that the method is considered to be "extensively evaluated across different tasks".
>
> We will provide explanations for the main points that you are concerned about.
>
> ---
> >*W1: I am wondering whether using such enhanced RM could improve the reasoning ability of existing LLMs.*
>
> Thanks for your insightful suggestions! Indeed, using an enhanced RM can improve the reasoning ability of existing LLMs. We have already conducted an experiment to validate this with a math comparison pair dataset (huggingface tag: ```reciprocate/math_dpo_pairs```). Specifically, we used this dataset during the second stage of GRAM training, with all baseline RMs also trained on the same dataset. We performed reward accuracy experiments on the corresponding test set and with best-of-n sampling on GSM8K, respectively. For the best-of-n sampling, we sample 16 outputs on LLaMA-3.1-8B-Instruct with 8-shot for each input.
>
> |Method|RM Accuracy|GSM8K (Best-of-n Sampling)|
> |:-|:-:|:-:|
> |LLaMA-3.1-8B-Instruct|-|83.9|
> |Discriminative RM|63.2|84.6|
> |Generative RM|61.3|85.5|
> |GRAM|**66.8**|**87.2**|
>
> The experimental results show that our method effectively improves RM accuracy on reasoning-related downstream tasks. Also, it further enhances the reasoning ability of LLMs with best-of-n sampling. We promise to include more experiments, including additional baselines and RL experiments, in the revised version to further verify the effectiveness of GRAM in improving reasoning abilities.
>
> >*W2: The influence of pre-training data on target domain should also be discussed.*
>
> In Figure 6 of the paper, we already show the impact of pre-training data on general domain adaptation. In response to your concern, we further conduct experiments in specific domains using LLaMA-3.2-3B-Instruct, along with 5k labeled summarization and harmlessness data.
>
> |Domain|Amount of Unlabeled Data|
> |:-|:-:|
> || 0k &nbsp; 100k &nbsp; 200k &nbsp; 400k &nbsp; 600k|
> |Summarization|56.5 &nbsp; 62.7 &nbsp; 66.3 &nbsp; 71.6 &nbsp; 73.4|
> |Harmlessness|58.4 &nbsp; 63.2 &nbsp; 65.9 &nbsp;  69.8 &nbsp; 72.1|
>
> The results from these experiments align with those in Figure 6. We see that as the amount of unlabeled data increases, the accuracy of GRAM generally improves for both models. This also highlights the crucial role of unlabeled data and the scaling effect on performance, suggesting that using larger unlabeled datasets can lead to better reward models. We promise to include these experiments in the revised version.
>
> >*Q1: Would the domain difference between pre-training and fine-tuning has a big impact on RM performance?*
>
> To respond to your concern, we conduct the following experiments on GRAM:
> - GRAM-v1: Pre-training on 100k unlabeled summarization response pairs (derived from TL;DR comparison data;huggingface tag: ```openai/summarize_from_feedback```), followed by fine-tuning on 5k labeled summarization data.
> - GRAM-v2: Pre-training on 100k general unlabeled data (including summarization responses), followed by fine-tuning on 5k labeled summarization data.
> - GRAM-v3: Pre-training on 100k general unlabeled data (without summarization-related data; specifically, we are using ChatGPT to filter out preference data related to summarization), followed by fine-tuning on 5k labelled summarization data.
>
> In these experiments, summarization is the downstream domain. GRAM-v1 uses pre-training data closest to summarization, fully utilizing summarization responses. GRAM-v2 uses general pre-training data, which might include some samples aligned with the downstream summarization task. GRAM-v3 uses pre-training data that is the most different from the summarization domain. Specifically, we use ChatGPT to exclude all samples related to summarization tasks. The results are as follows:
>
> |Method|Accuracy (Summarization)|
> |:-|:-:|
> |RM w/o Pre-training |56.5|
> |GRAM-v1|**74.7**|
> |GRAM-v2|71.6|
> |GRAM-v3|67.4|
>
> The experimental results demonstrate that pre-training data more closely aligned with the target domain results in better performance in that domain. In fact, this experimental observation is consistent with the common practice in LLMs, where incorporating as much domain-specific data as possible during pre-training typically leads to better performance on downstream tasks. Additionally, our results show that the pre-training approach exhibits strong robustness—despite significant domain differences, it still yields positive contributions to performance.
>
> We appreciate your valuable suggestion and promise to include more discussion and experiments in the revised version.
>
> ---
> We sincerely thank you for your positive feedback on our paper!
> Thank you once again for your time.
>
> Best,
>
> Authors

---

### Official Review · Reviewer_WL65 · 2025-03-12

**Overall Recommendation:** 3

**Summary:**

Authors propose improvements on the training of generative reward models (GenRMs). First, they pre-train GenRMs on pairs of responses. Second, they apply label smoothing. This approach is called GRAM. Authors also make an observation that label smoothing shall be understood as the regularization of Bradley-Terry scores. Authors experimentally validate their approach with UnifiedFeedback and Llama-3.1-8B and 3.2-3B instruct models. When trained on the same 400k preference data, proposed GRAM approach outperforms generative and discriminative RM baselines. When applied to Best-of-n sampling, GRAM is also shown to generalize better on AlpacaEval. When fine-tuned on specific domains (summarization and helpfulness), GRAM is also more sample-efficient than baseline methods.

## update after author rebuttal

I will maintain my score. Authors answered my clarifying questions very clearly, but I expected them to be answered in my original score. I still feel this is a solidly executed yet somewhat incremental paper, hence I would be happy to see it accepted, yet open to change my opinion.

**Claims And Evidence:**

The main claim of this paper is that GRAM generalizes better than previous reward modeling approaches. Across three different use-cases of RMs - preference ranking, best-of-n sampling, task-specific fine-tuning - the sample efficiency of GRAM is consistently shown. Therefore, the claim is empirically well supported.

**Essential References Not Discussed:**

AlpacaFarm paper https://arxiv.org/abs/2305.14387 is cited but their use of label smoothing was not discussed. AlpacaFarm paper considered it as synthetic label noise.

Label smoothing wasn't cited. Although it's well-established, it's worth citation: Szegedy et al https://arxiv.org/abs/1512.00567

Generative Reward Models (Mahan et al, 2024 https://arxiv.org/abs/2410.12832) paper has already made an observation that GenRMs generalize better on out-of-domain. I understand the paper was rejected from ICLR but it is worth citation and discussion. Just to be sure, I didn't discount the contribution of authors' paper because of Mahan et al because it's not yet accepted.

**Experimental Designs Or Analyses:**

I checked experimental designs and analyses for all experiments in the main paper and Appendix Figure 9, 10. As I discussed in 'Methods And Evaluation Criteria', there is some room for improvements, but they are not critical.

**Methods And Evaluation Criteria:**

The proposed two extensions of GenRM are standard modeling techniques and hence they are appropriate. The first approach of pre-training on response pairs can be understood as continued pre-training on the same domain of unlabeled data; this is a standard, well-established technique originating from AdaptaBERT https://aclanthology.org/D19-1433/ . The second, application of label smoothing is also standard in the training of classification models (from Szgedy et al https://arxiv.org/abs/1512.00567 ) and already explored in reward modeling context in the AlpacaFarm paper https://arxiv.org/abs/2305.14387 . Also, the concept of reward model pretraining was considered in Bai et al (2022) https://arxiv.org/abs/2204.05862 although in a different form. Hence these techniques are all well-established, principled methods.

The evaluation criteria are also standard. RewardBench is well-established benchmark for evaluating generic-purpose reward models, although there are new ones such as RM-Bench, JudgeBench, FollowBenchEval. See Saha et al. https://arxiv.org/abs/2501.18099 for their experimental setting. Evaluation of Best-of-N on AlpacaEval is acceptable but the ideal setup would be to use the standard GPT-based evaluator from AlpacaEval itself rather than using a proxy reward model. Fine-tuning experiments also use standard domain-specific preference datasets, which is good.

**Other Comments Or Suggestions:**

- Line 047 column 2: the systems cannot directly generate from text their own supervision signals -> There actually are papers on this type of approach, hence the statement is a bit unfair to them. For ex Self-Rewarding Language Models https://arxiv.org/abs/2401.10020 .
- Line 028 column 2: Lee et al reference doesn't have year information

**Other Strengths And Weaknesses:**

Experimental analyses are very comprehensive, meticulously studying each of the design choices, for example Table 3/Figure 9 and also studying scaling aspects (Figure 6). The strength of the proposed method is also consistent across experiments.

On the other hand, I find technical contribution of the paper is a bit simple - pre-training on pairs of responses and label smoothing. These are already very well-established concepts in machine learning, hence the contribution of this paper is mostly empirical, and maybe better suited at NLP conferences than ML conferences.

**Questions For Authors:**

On which dataset do authors run their stage 1 (task 1 pre-training)? I suspect they just used responses from UnifiedFeedback but I wasn't able to find an explicit discussion of the data used for stage 1 training.

In Figure 5, do RM fine-tuned with G/D-baselines mean RMs trained on UnifiedFeedback?

**Relation To Broader Scientific Literature:**

Connections to previous work in literature I already made in 'Methods And Evaluation Criteria' section are notable. The idea of pre-training reward models was explored in Bai et al (2022) https://arxiv.org/abs/2204.05862, but they focused on sourcing preference data from web. Authors' approach has the benefit of leveraging unlabeled data. However, in the experiments, I believe they pre-train only on UnifiedFeedback data, which are labeled.

This approach can also be understood as a form of unsupervised domain adaptation https://aclanthology.org/D19-1433/ , and the connection may foster other ideas, techniques, and theory from domain adaptation to apply to reward modeling.

**Theoretical Claims:**

I checked the correctness of the regularization characterization of label smoothing for Bradley-Terry models. I checked steps in Appendix D.2, which are straightforward algebra.

---

> ### Author Rebuttal · Authors · 2025-03-31
>
> Dear Reviewer WL65,
>
> We appreciate the reviewer’s constructive and thoughtful feedback.
> We appreciate your recognition that the main claim of our paper is "empirically well supported", and that our experiments are "comprehensive, meticulously studying each of the design choices".
>
> We provide explanations for the main points that you are concerned about.
>
> ---
>
> >*W1: The technical contribution seems simple—pre-training and label smoothing—both well-established in ML. The contribution is empirical and may be more suited for NLP conferences than ML.*
>
> Thanks for your insightful comment! As you said "pre-training and label smoothing are well-established concepts in ML", we would like to clarify that our contribution is not about innovating them, but rather introducing these concepts to achieve superior reward modeling. The rationale behind this is as follows:
>
> - While applying reward models to align LLMs is a compelling direction, training these models still heavily relies on labeled data. We expect to enhance reward modeling by pre-training on unlabeled data to reduce dependence on labeled data. However, this pre-training approach has never been explored and presents significant challenges. The difficulty arises from that, unlike self-supervised approaches, systems cannot directly generate their own supervision signals from text for training reward models. In this work, we thoroughly explore the use of unlabeled data in reward modeling and propose a specific pre-training reward model procedure. We also discuss its effectiveness from both theoretical and practical perspectives.
>
> - Although label smoothing has been shown to be effective in many tasks, it has not been well-established in reward modeling. In this work, we theoretically demonstrate that the training objective of generative reward models can be reformulated into a more elegant form: we are essentially optimizing the Bradley-Terry model with modified label smoothing. This result is significant as it establishes a connection between discriminative and generative reward modeling methods—both of which fundamentally train LLMs to perform pairwise ranking, thereby pushing the boundaries of reward modeling and directly contributing to improved generalization.
>
> >*Q1: On which dataset do authors run their stage 1 (task 1 pre-training)? The similar idea of pre-training reward models is mentioned in Bai et al. (2022).*
>
> The pre-training responses also come from Unified Feedback. Please note that while this data includes preference labels, we do not use these labels in our pre-training process. Instead, we only use the response pairs to simulate unlabeled data and validate the effectiveness of our method.
>
> Since our pre-training approach only uses responses for unsupervised training (which can be directly sampled from LLMs), it allows for easier scalability. We also present two key insights from our experiments for pre-training. First, as shown in Figure 5, during downstream adaptation, our pre-training effectively achieves generalization and reduces the need for specific labeled data. For example, in the summarization task, we achieve performance comparable to training from scratch with approximately 100k labeled data using only 5k data points. Second, as demonstrated in Section 5.1, we find that the more unlabeled data used in training, the greater the benefit to downstream performance, regardless of whether the downstream labeled data is small or large. Building on these insights, in real-world reward model training, we could first collect many responses using cost-effective methods, such as sampling from LLMs, and then fine-tune with a small amount of labeled preference data.
>
> In contrast, Bai et al. (2022) propose a pre-training approach that uses large, labeled preference data sets for supervised training, followed by domain-specific fine-tuning. This method is difficult to scale as it still relies heavily on labeled preference data, which is often scarce in real-world scenarios.
>
> Thank you for your helpful suggestion! In the revised version, we promise to provide a more detailed description and experiment-based analysis for your concern.
>
> >*Q2: In Figure 5, do RM fine-tuned with G/D-baselines mean RMs trained on UnifiedFeedback?*
>
> Yes, the data used to fine-tune the RMs for forming the G/D-baselines comes from UnifiedFeedback. Note that this data is the same as the one used by GRAM in the fine-tuning stage, which includes preference annotations.
>
> >*S1: Insufficient description of related work and errors in reference presentation.*
>
> Thank you for your helpful feedback! We promise to correct the description and presentation of the related work in the revised version.
>
> >*S2: Essential references are not discussed.*
>
> Thanks for your valuable suggestion! We promise to include the discussion of the essential references in the revised version.
>
> ---
> We sincerely appreciate your positive feedback and thank you again for your time.
>
> Best,
>
> Authors

---

> > ### Comment · Reviewer_WL65 · 2025-04-02
> >
> > I recognize that I undervalued the paper's methodological contributions, hence I increased my score accordingly.
> >
> > >The pre-training responses also come from Unified Feedback. Please note that while this data includes preference labels, we do not use these labels in our pre-training process. Instead, we only use the response pairs to simulate unlabeled data and validate the effectiveness of our method.
> >
> > I understand this, and I understand the convenience of this setup, but this point could've been much stronger if authors leveraged organically unlabeled, large-scale data to show this point. This setup is a bit artificial since the original data, including prompts & responses were already curated to be useful for the supervised setting, just that labels are hidden for pre-training.

---

> > > ### Author Response · Authors · 2025-04-03
> > >
> > > Thank you for your valuable feedback and active engagement during the rebuttal process. We appreciate your insightful suggestion regarding the reconstruction of our pre-training data setup. We promise to include a discussion of this aspect in the revised version. Thank you once again for your endorsement!

---

### Official Review · Reviewer_3tRc · 2025-03-13

**Overall Recommendation:** 3

**Summary:**

This work introduces GRAM, a generative foundation reward model for aligning LLMs with human preferences. Unlike conventional reward models that rely only on labeled human preference data, GRAM incorporates both labeled and unlabeled data through a two-stage training process: unsupervised “pre-training” on input-response pairs, followed by fine-tuning with human preference data for task-specific alignment. The authors demonstrate that the incorporation of label smoothing unifies generative and discriminative reward models under a shared training objective. The authors demonstrate the effectiveness of their proposed approach by first pre-training on 400,000 examples Extensive evaluations across response ranking, reinforcement learning from human feedback (RLHF), and task adaptation show that GRAM generalizes effectively across tasks, achieving improvements over baselines.

**Claims And Evidence:**

The claims about GRAMs performance are supported by their experiments. In particular, the method leads to a reward model that can generalize to out-of-distribution data, and can be used in multiple settings (as both a pairwise reward model and list-wise reward model). Some of the experimentation details are not clear, which, if improved, will make the evidence even more convincing.

**Essential References Not Discussed:**

The following works both propose a variant of a generative reward model. The main difference that I can see with what is done here is that both these works allow the model to reason over the 2 preference options prior to providing its judgement. They are similar enough to the current work that readers should know that they exist, as they can provide a direction for future works.
- Ankner et al., 2024. Critique-out-Loud Reward Models
- Mahan et al., 2024. Generative Reward Models

**Experimental Designs Or Analyses:**

- The experimental design stands to be improved slightly. For example, the authors describe a version of label smoothing used for their method, while the ablation on baselines uses a different version of label smoothing. This difference of methods should either be explained, or rectified to use the same version of label smoothing.
- Additionally, some of the details in the experiment on list-wise response ranking are unclear. What is the purpose of comparing with a proxy model that is trained on less data? Additionally, which model is used as the policy to generate the responses being ranked?

**Methods And Evaluation Criteria:**

One component of GRAM that does not make sense for this problem is the pre-training stage, where the model learns to generate 2 responses to a prompt. Theoretically, it’s not clear to me why the unsupervised learning portion should be effective (although the experiments show it to be, specifically in Section 5.1). In particular, given that the reward model is never asked to generate responses, and the responses at test time are likely to come from a different distribution than the training data, the reason this is beneficial is unclear to me. This is one area that could use some explanation or exploration.

The evaluations make sense for the problem at hand though.

**Other Comments Or Suggestions:**

Some of the experimental details need to be clarified. See the comments in “Experimental Designs Or Analyses” and “Questions for Authors”.

**Other Strengths And Weaknesses:**

Strengths:
- The results are quite good compared to baselines.
- This version of pre-training for reward modeling is quite novel, to my knowledge.
Weaknesses:
- The experimental setup is unclear. For example, in Table 1, what are the methods denoted (baseline)?
- The main set of experiments seem to use different versions of label smoothing for the baseline vs. the full method. Because we only see the label smoothing + pretraining together, this makes it hard to determine the effects of each individual component

**Questions For Authors:**

- Some aspects of the experimental setup are unclear to me. Are there 2 splits of 400,000 examples? One split used for pre-training and the other for fine-tuning?
- Where do the pretraining responses come from (y1 and y2)? Given that the training dataset contains pairwise examples, do you use those for the pre-training stage, or do you actually generate the responses with the models you will train (Llama-3.1-8B and Llama-3.2-3B)? In general, are the responses that are used for pre-training generated by the same model that you will then train with that data?
- Starting at line 126, “When applying this model to score a new input-response”, you suggest that you generate a reference response with the current LLM. I am unfamiliar with this method of scoring a single response. Has this method previously been used in other works? In particular, one concern I have with this method is that it may lead to vastly different scores depending on which LLM generates the reference response.
- Why is the “label smoothing” applied to the models in your experiments different from the method that you proposed in Section 3.3?

**Relation To Broader Scientific Literature:**

I think the use of generative reward models has become a very popular idea in the past 6-9 months and the community may appreciate this work. In particular, the idea of performing unsupervised training specifically for reward modelling is novel, however, I think the exact setup of unsupervised learning is not well motivated and unclear.

**Theoretical Claims:**

I did not verify the derivation of equation 8.

---

> ### Author Rebuttal · Authors · 2025-03-31
>
> Dear Reviewer 3tRc,
>
> We would like to thank the reviewer for the positive feedback regarding the novelty of pre-training for reward models and the strong results.
> Below, we explain the main points you are concerned about.
>
> ---
>
> >*One component that does not make sense for this problem is the pre-training stage.*
>
> We would like to clarify the motivation behind our pre-training design. As discussed in Section Appendix A, we can understand our method from a feature learning perspective. In generative reward modeling, the loss function can be given by
> $$
> L\_{\mathrm{g}}(\theta) = L\_{\mathrm{gp}}(\theta\_{\mathrm{gp}})+L\_{\mathrm{gf}}(\theta)
> $$
> The feature optimization term $L\_{\mathrm{gf}}(\theta)$ is implicitly defined and optimized by adjusting preference generation. Traditional generative reward modeling optimizes these objectives using costly preference labels. This work explores a pre-training approach to optimize $L\_{\mathrm{gf}}(\theta)$ first.
>
> To this end, we propose using an auxiliary task to optimize these features without relying on labeled preferences. Specifically, we utilize conditional probabilities to characterize interrelationships, i.e., $-\log \pi\_{\theta}(y\_{b}|x,y_a)$. Besides, considering a distributional error arising from both $\pi\_{\theta}$ and the $y_a$ or $y_b$ to a given input $x$, we introduce an SFT loss as a regularization term, i.e., $-\log \pi\_{\theta}(y|x)$. After careful derivation, we obtain
> $$
> L\_{\mathrm{gf}}(\theta) = -\mathbb{E}\_{(x, y_a, y_b) \sim D_u} [ \log \pi_\theta([y_a, y_b] | x) ]
> $$
> This predicts two responses. The model can better generalize and improve its reward capability by focusing on learning meaningful features rather than surface-level ones (e.g., length). Also, this pre-training strategy has been highlighted to make sense by Reviewer Jdyw. Thanks for the feedback!
>
> >*What is the purpose of comparing with a proxy model trained on less data? Which model is used as the policy for generating ranked responses?*
>
> Training a proxy model tests the method's modeling capability. A good proxy score indicates strong performance on ID, reflecting good modeling. Our goal is to design a method with both strong modeling and generalization. Thus, testing with the proxy score is crucial. This setup is also supported by Yang et al. (2024).
>
> As for the sampling policy, we use the LLaMA-3.1-8B model, as described in Section 4.4 (line 319).
>
> >*W1: In Table 1, what are the methods denoted (baseline)?*
>
> Our main baselines include several methods (such as Freeze and Regularization) aimed at enhancing generalization and discriminative models to demonstrate the promising generalization of our GRAM.
>
> >*W2&Q4: Lack of detailed description on the baseline version of label smoothing (LS).*
>
> The LS we use differs from the standard LS. Our method applies LS to all candidate label tokens, as described in Section 3.3, whereas the traditional method applies it to all tokens in the vocabulary. We theoretically demonstrate that our method is more effective, potentially optimizing with a constrained Bradley-Terry model. Apart from the explanation in Appendix D.2, we have conducted further experiments on the LLaMA-3.2-3B-Instruct to explore this in more detail below.
>
> |Method|UniFeed(ID)|RewardBench(OOD)|
> |:-|:-:|:-:|
> |GRAM w/o LS|68.4|82.1|
> |GRAM w/ Our LS|**70.6**|**83.6**|
> |GRAM w/ Standard LS|69.1|82.5|
> |G-Baseline w/ Our LS|66.7|80.2|
> |G-Baseline w/ Standard LS|66.2|79.0|
>
> As the experimental results show, our LS leads to better reward modeling compared to the standard LS. These results also show the role of LS in training GRAM. We promise to add more discussions on this in the revised version.
>
> >*Q1&Q2: Are there two 400,000-example splits, one for pre-training and one for fine-tuning? Can we collect pre-training data by sampling from LLaMA models?*
>
> Yes, both splits of 400,000 examples are randomly selected from the Unified Feedback dataset. The pre-training responses also come from Unified Feedback, but we only use the responses without their preference labels to simulate unlabeled responses. Sampling responses from LLaMA to obtain unlabeled data is a good idea, and we commit to exploring this method further. This work focused on the time-consuming nature of sampling large-scale labeled response pairs. Also, we expect such available data to exist for direct training in real-world scenarios. Thus, we randomly select response pairs from Unified Feedback to simulate available labeled response pairs.
>
> >*Q3: This scoring by Eq. 3 may yield vastly different scores depending on the reference response.*
>
> This setup is reasonable. In BoN sampling, we keep the same reference for all candidate outputs, ensuring the scores are comparable. In RL, the recent work *Remax* uses the difference from a reference as the reward and interprets its reasonableness from a reward baseline perspective.
>
> ---
> We sincerely thank you for your positive feedback on our paper!
>
> Best,
>
> Authors

---

### Decision · Program_Chairs · 2025-05-01

**Decision:**

Accept (poster)

**Comment:**

This paper proposes to train a (Generative) Reward Model in two stages: first a pre-training stage (where the model is trained to generate pairs of responses, without preference labels), followed by a fine-tuning stage on labeled preferences (also leveraging label smoothing). This is shown to outperform discriminative Reward Models in the large majority of empirical results.

Overall Reviewers would like to see the paper accepted, but without considering it a must-have (all 3's). I personally find this work intriguing, as it can be surprising to see the benefits brought by training the RM to generate responses during this "pre-training" stage. Although I feel like the "why this works" may not be fully understood, I find the empirical evaluation to be extensive enough to justify the method and deserve being shared with the ICML community. The pre-training of RMs is an under-studied topic and this paper makes it clear that it's worth looking into it more closely.